**Secondary aerosol formation under a special dust transport**
**event: impacts from unusually enhanced ozone and dust**
**backflows over the ocean**
Da Lu[1], Hao Li[1], Guochen Wang[1], Mengke Tian[1], Xiaofei Qin[1], Na Zhao[1], Juntao
Huo[2], Fan Yang[3], Yanfen Lin[2], Jia Chen[2], Qingyan Fu[2], Yusen Duan[2], Xinyi Dong[4],
Congrui Deng[1], Sabur F. Abdullaev[5], Kan Huang[1,6*]
[1]Center for Atmospheric Chemistry Study, Shanghai Key Laboratory of Atmospheric
Particle Pollution and Prevention (LAP[3]), National Observations and Research Station
for Wetland Ecosystems of the Yangtze Estuary, Department of Environmental
Science and Engineering, Fudan University, Shanghai, 200433, China
[2]State Ecologic Environmental Scientific Observation and Research Station for
Dianshan Lake, Shanghai Environmental Monitoring Center, Shanghai, 200030,
China
[3]Pudong New District Environmental Monitoring Station, Shanghai 200122, China
[4]School of Atmospheric Sciences, Nanjing University, Nanjing 210023, China
[5]Physical Technical Institute of the Academy of Sciences of Tajikistan, Dushanbe,
Tajikistan
[6]Institute of Eco-Chongming (IEC), Shanghai, 202162, China
Corresponding author: huangkan@fudan.edu.cn
**Abstract**
In the autumn of 2019, a five-day long-lasting dust event was observed using a
synergy of field measurement techniques in Shanghai. This particular dust event stood

out from others due to its unique characteristics, including low wind speed, high relative humidity, elevated levels of gaseous precursors, and contrasting wind patterns at different altitudes. During this event, three distinct dust stages were identified. The first stage was a typical dust invasion characterized by high concentrations of particulate matters but relatively short duration. In contrast, the second stage exhibited an unusual enhancement of ozone, attributed to compound causes of weak synoptic system, transport from the ocean, and subsidence of high-altitude ozone down drafted by dust. Consequently, gas phase oxidation served as the major formation pathway of sulfate and nitrate. In the third stage of dust, a noteworthy phenomenon known as dust backflow occurred. The dust plume originated from the Shandong Peninsula and slowly drifted over the Yellow Sea and the East China Sea before eventually returning to Shanghai. Evidence of this backflow was found through the enrichment of marine vessel emissions (V and Ni) and increased solubility of calcium. Under the influence of humid oceanic breezes, the formation of nitrate was dominated by aqueous processing. Additionally, part of nitrate and sulfate were directly transported via sea salts, evidenced by their co-variation with $Na^+$ and confirmed through thermodynamic modeling. The uptake of $NH_3$ on particles, influenced by the contributions of alkali metal ions and aerosol pH, regulated the formation potential of secondary aerosol. By developing an upstream-receptor relationship method, the amounts of transported and secondarily formed aerosol species were separated. This study highlights that the transport pathway of dust, coupled with environmental conditions, can significantly modify the aerosol properties, especially at the complex land-sea interface.

## 1. Introduction

Dust serves as a significant natural source of aerosols, constituting approximately
half of the tropospheric aerosols (Zheng et al., 2016). Dust aerosols play crucial roles
in environmental and climatic changes by affecting the radiation balance (Feng et al.,
2020; Nagashima et al., 2016; Goodman et al., 2019). The optical properties of dust
aerosols are influenced by various parameters of iron oxides, including refractive
indices, size distributions, and mineralogical compositions. Consequently, these factors
introduce potential uncertainties regarding the role of dust in climate forcing (Zhang et
al., 2015; Jeong, 2008). Furthermore, dust aerosols have important impacts on
tropospheric chemistry by participating in heterogeneous and photolysis reactions in
the atmosphere (Wang et al., 2014; Liu et al., 2018). During transport, dust can mix
with gaseous pollutants, toxic metals, and soot, thereby affecting air quality
immediately and potentially posing public health hazards (Liu et al., 2021; Wang et al.,
2021). Moreover, Barkley et al. (2021) found that iron-containing aerosols transported
from Africa to the equatorial North Atlantic Ocean provided plentiful nutrients to algae
in the ocean and accumulated inside algae.
The irregular shapes of dust particles provide an efficient medium for
heterogeneous reactions with $NO_2$, $O_3$, $SO_2$, and $NH_3$, thereby altering the particle size
spectrum, hygroscopicity, and radiative properties (Hsu et al., 2014; Tian et al., 2021;
Jiang et al., 2018). Jiang et al. (2018) observed a significant increase in nitrate and
sulfate concentrations during a dust period in March 2010 in Shanghai. This elevation
was attributed to the presence of moderate to high levels of relative humidity and
gaseous precursors, implying that dust can efficiently promote the formation of sulfate
and nitrate. Previous studies have revealed that $HNO_3$ formed through the reactions of
$NO_2$ with hydroxyl radical or $N_2O_5$ hydrolysis preferentially reacts with mineral dust
particles and produce nitrate, which serves as the primary source of nitrate during dust
episodes (Tang et al., 2016; Wu et al., 2020). Improvements in the simulation of sulfate
were achieved by employing various parameterization schemes for the heterogeneous
uptake of $SO_2$ on natural dust surfaces in the presence of $NH_3$ and $NO_2$ under different
relative humidity conditions(Zhang et al., 2019). Wang et al. (2018) simulated that
heterogeneous reactions on dust accounted for the majority of nitrate over the Yellow
Sea and the East China Sea during the dust long-range transport. Tang et al. (2017)
conducted a comprehensive review on the effect of dust heterogeneous reactions on the
tropospheric oxidation capacity. They proposed that high RH ($> 80\%$) and a wider range
of temperature should be considered in the laboratory studies of heterogeneous
reactions of mineral dust. Additionally, more comprehensive kinetic models should be
developed to understand the complex multiphase reactions.

Controversies have arisen regarding the mixing of dust and anthropogenic aerosols.

Zhang et al. (2005) found that anthropogenic aerosols separated with dust during a dust
event in Qingdao, China. Coincidentally, a time-lag between dust and anthropogenic
aerosols was observed in Japan and South Korea downstream of the dust transport.
Single particle analysis revealed that sulfate in fine particles appeared 12 hours before
the dust arrival in Japan. Wang et al. (2013) also observed a lag of 10 - 12 hours between
dust and anthropogenic aerosols on a dust day in Shanghai (Wang et al., 2013).
Furthermore, Huang et al. (2019) documented vertical differences in long-transported
aerosols during a pollution event in Taiwan. Dust from the Gobi Desert in Inner
Mongolia and China existed at the altitudes of 0.8km and 1.90km, respectively, while
biomass burning aerosols from South Asia were present at higher altitudes of 3.5km.
Coastal regions often experience a mixture of inland anthropogenic emissions and
releases from the ocean, making regional pollution complex in these areas (Wang et
al.; Hilario et al., 2020; Patel and Rastogi, 2020; Perez et al., 2016; Wang et al., 2017).
The eastern coast of China,bordering the East China Sea and the Yellow Sea, is
particularly influenced by the Asian monsoon and high emissions from inland industries,
resulting in highly intricate meteorological and pollution conditions (Hilario et al.,
2020). Furthermore, the marine boundary layer in this region exhibits significant
seasonal and diurnal variations in , relative humidity and temperature further impacting
photochemical processes and heterogeneous reactions on aerosol surfaces (Zhao et al.,
2021). Sea and land breezes play a crucial role in this coastal area. During the night,
land breezes carry pollutants from the land to the sea. Subsequently, during the day,
these land breezes transform into sea breezes, bringing the pollutants back over the sea.
This phenomenon leads to an increase in air pollutants over the land (Zhao et al., 2021).
For instance, Wang et al. (2022b) found that during the ozone pollution in Shanghai in
2018, the presence of $O_3$ at high altitudes at night was transported vertically downward
during the daytime and high $O_3$ over the ocean was transported horizontally to the land,
jointly contributing to regional $O_3$ pollution in Shanghai. Also, one dust episode in 2014
was observed over Shanghai via detouring from northern China due to the blocked north
Pacific subtropical high-pressure system (Wang et al., 2018).
Previous studies have shown that about 70% of Asian dust traverses the eastern
coast of China before moving towards the Korean Peninsula, the Sea of Japan, and
eventually reaching the Pacific Ocean. The eastern coast of China serves as a crucial
route for Asian dust transport to the Pacific Ocean (Arimoto et al., 1997; Huang et al.,
2010). Most previous research has focused on typical dust events characterized by
strong intensities, high wind speed, low humidity, and low oxidants (Li et al., 2017; Ma
et al., 2019; Xu et al., 2017; Xie et al., 2005). In contrast, this study aims to depict an
atypical dust event that was observed in Shanghai, a coastal mega-city in Eastern China.
The unusualness of the meteorological conditions, transport pathways, and air
pollutants during the particular dust event was explicitly described. The study involves
categorizing the dust event into three stages and comparing the aerosol chemical
compositions between these stages. By focusing on the second and third stages, the
different formation mechanisms of nitrate and sulfate were investigated. The amounts
of major aerosol species from transport and secondary formation were estimated based
on a simplified method of relating the upstream and receptor simultaneous
measurements.

**2. Methodology**
**2.1. Observational sites**
At Shanghai Pudong Environmental Monitoring Station (31°13′ N, 121°32′E),
comprehensive measurements of various atmospheric parameters were conducted.
All the instruments were installed on the top floor of a building, about 25m above
the ground level. As shown in Figure S1, the sampling site is situated at the eastern
tip of Shanghai, close to the coastal line. During November, the mean temperature
and relative humidity in Shanghai were recorded as 17.3°C and 72% respectively.
In autumn and winter, air pollutants originating from upstream urban regions often
undergo transport to Shanghai via high-pressure systems. Additionally, air
pollutants in Shanghai tended to linger at the sea/land boundary regions due to the
sea-land breeze (Shen et al., 2019).
In addition to the measurements taken in Shanghai, data from environmental
monitoring stations in Qingdao and Lianyungang are also incorporated into this
study.

**2.2. Instrumentation**
A set of online instruments was set up at the Pudong observational site. Inorganic ions
($NO_3^-$, $SO_4^{2-}$, $Cl^-$, $Na^+$, $NH_4^+$, $K^+$, $Mg^{2+}$, $Ca^{2+}$) in $PM_{2.5}$ and soluble gases ($NH_3$, $HNO_3$,
HCl, HONO) were measured by an online ion chromatography (IC, MARGA-1S,
Metrohm). It operated at a flow rate of 16.7 L/min with a time resolution of one hour.
Briefly, air was drawn into a $PM_{2.5}$ cyclone inlet and passed through a wet rotating
denuder (gases) and a steam jet aerosol collector (aerosols). Subsequently, the aqueous
samples were analyzed with ion chromatography. More details can be found in (Xu et
al., 2020). Hourly trace metals (Si, Ca, Cu, Fe, K, Co, Mn, Cr, Zn, Pb, As, Cd, V, Ni) in
$PM_{2.5}$ were measured by using the Xact 625 multi-metals monitor (Cooper
Environmental, Beaverton, OR, USA). Particles were collected onto a Teflon filter tape
at a flow rate of 16.7 L/min, and then transported into the spectrometer where the
particles were analyzed with an X-ray fluorescence. Organic carbon and elemental
carbon were measured by an in situ Semi-Continuous Organic Carbon and Elemental
Carbon aerosol analyzer (RT-4, Sunset Laboratory, Beaverton, Oregon, USA). Samples
were collected for 40 min and then analyzed in the following 20 min. The concentration
of mineral aerosols is calculated by summing the major mineral elements with oxygen
for their normal oxides, i.e., [Minerals]= (2.2*Al+2.49*Si+1.63*Ca+2.42*Fe+1.94*Ti)
(Malm et al., 1994). The concentration of OM (organic matters) is estimated by
multiplying OC with a factor of 2.

168  The concentrations of particles and gaseous pollutants were measured by a set of

169 Thermo Fisher Scientific instruments, including $PM_{2.5}$ (Thermo 5030i), $PM_{10}$ (Thermo

170 5030i), $SO_2$ (Thermo Fisher 43i), $NO_x$ (Thermo Fisher 42i), $O_3$ (Thermo Fisher 49i),

171 and CO (Thermo Fisher 48i-TLE). These parameters were measured at the temporal

172 resolution of 5min. Meteorological parameters (ambient temperature, relative humidity,

173 wind speed, and wind direction) were obtained by a Vaisala Weather transmitter

174 (WXT520) at the temporal resolution of 1min. The height of planetary boundary layer

175 (PBL) was retrieved from a ceilometer (CL31, Vaisala) at the temporal resolution of 30

176 min. Vertical profiles of aerosol optical properties were obtained by an aerosol lidar

177 (AGJ, AIOFM)at the temporal resolution of 30 min and vertical resolution of 7.5 m,

178 respectively. Vertical profiles of ozone were obtained by an ozone lidar (LIDAR-G-

179 2000, WUXIZHONGKE) at the temporal resolution of 15 min and vertical resolution

180 of 7.5 m, respectively. All instruments are routinely maintained and calibrated to ensure

181 the quality of data.

183 **2.3. Thermodynamic simulation of aerosol pH and aerosol liquid water content**

184  The ISORROPIA II model is subject to the principle of minimizing the Gibbs energy

185 of the multi-phase aerosol system, leading to a computationally intensive optimization

186 problem (Song et al., 2018). The model can predict the physical state and compositions

187 of atmospheric inorganic species ($NH_4^+$, $Na^+$, $K^+$, $Mg^{2+}$, $Ca^{2+}$, $SO_4^{2-}$, $NO_3^-$ and $Cl^-$)

188 with their gas- and particle-phase concentrations and meteorological parameters

189 (relative humidity and temperature) as model inputs. The model includes two modes,

190 i.e., reverse and forward mode. The reverse mode calculates the equilibrium

191 partitioning based on aerosol-phase concentrations only, while the latter uses both

aerosol-phase and gas-phase concentrations as inputs. Moreover, particles can be
assumed as "metastable" with liquid-phase but no solid participating while "stable"
with the liquid and solid phases or both. The ISORROPIA running in the forward mode
at the metastable state was applied in this study. Aerosol pH was calculated based on
the equilibrium particle hydronium ion concentration and aerosol liquid water content
(ALWC) obtained from model results. The performances and advantages of
ISORROPIA over the usage of other thermodynamic equilibrium codes has been
assessed in numerous studies (Nenes et al., 1998; West et al., 1999; Ansari and Pandis,
1999; Yu et al., 2005).

**2.4. Hybrid Single-Particle Lagrangian Integrated Trajectory Model**
The HYSPLIT (Hybrid Single-Particle Lagrangian Integrated Trajectory) was
used to compute the backward trajectories of the air parcels during the dust events. It is
a widely used model that computes dispersion following the particle or puff. The
advection of a particle or puff is computed from the average of the three-dimensional
velocity vectors for the initial position and the first-guess position (Draxler
and Hess, 1998). Turbulent velocity components are expressed as a function of the
velocity variance, a statistical quantity derived from the meteorological data, and the
Lagrangian time scale. The calculation of air mass trajectories can be used to depict the
airflow patterns for interpreting the transport of air pollutants over various spatial and
temporal ranges (Stein et al., 2015). In this study, the HYSPLIT model was driven by
meteorological data outputs from the Global Data Assimilation System (GDAS) (Su et
al., 2015), which is available at ftp://arlftp.arlhq.noaa.gov/pub/archives/gdas1. Air mass
trajectories were launched at different heights from the ground and a total duration of
48 hours simulation was conducted.

**2.5. Calculation of uptake coefficient of NH₃ ($\gamma$ NH3) on particles**

NH₃, being the most abundant alkaline species in the atmosphere, plays a crucial
role in acid neutralization and secondary aerosol formation. To assess the gas-particle
partitioning of NH₃, the uptake coefficient of NH₃ ($\gamma_{NH3}$) on particles is calculated as
below. Initially, the quasi-first-order reaction rate constant for heterogeneous
conversion from NH₃ to NH₄⁺ ($k_{het}$, s⁻¹) is calculated is calculated by Eq. (1) (Liu et
al., 2022).
$$k_{het} = \frac{2(C_{NH_4^+,t_2} - C_{NH_4^+,t_1})}{(C_{NH_3,t_2} + C_{NH_3,t_1})(t_2 - t_1)} \qquad (1)$$

$k_{het}$ is only valid when $c_{NH4+}$ increases, while $c_{NH3}$ decreases assuming a constant
emission rate from $t_1$ to $t_2$ (1 h in this study). Then, the uptake coefficient of NH₃ ($\gamma_{NH3}$)
on particles can be calculated as below (Liu et al., 2022; Wang and Lu, 2016).
$$\gamma_{NH_3} = \frac{4k_{het}}{S\omega} = \frac{4k_{het}}{S\sqrt{\frac{8RT}{\pi M}}} \qquad (2)$$

where S is the surface area of particles (m² m⁻³) measured using SMPS and APS. $\omega$ is
the velocity of NH₃ molecules. T is the ambient temperature (K). R is the ideal gas
constant, and M is the molecular weight of NH₃ (kg mol⁻¹).

**3. Results and Discussion**

**3.1. Characteristics of an unusual dust event**

Figure 1 shows the time series of PM₁₀, PM₂.₅, meteorological parameters, as well
as the vertical profiles of aerosol extinction coefficient and depolarization ratio
observed at the Shanghai sampling site from October 25 to November 6, 2019. From
October 25 to 28, the mean wind speed was 0.9±0.72m/s with a peak value of 3.1m/s,
remaining relatively low, and predominantly blowing from the northwest. The mean
concentrations of $PM_{2.5}$ and $PM_{10}$ were 34.7 and 44.2 μg/m$^3$, respectively. Starting at
4:00 LST on October 29, the concentration of $PM_{10}$ increased sharply and lasted till
November 2 (Figure 1d). The aerosol lidar observation indicated that both the aerosol
extinction coefficient and depolarization ratio extended from the ground to around 2km
during the same period. In general, if the particle depolarization ratio exceeds 10%, the
aerosol is identified as mineral dust (Shimizu et al., 2004) due to the nonsphericity
(irregular shapes) and relatively large size of particles (Mcneil and Carswell, 1975).
Notably, the enhanced depolarization ratio (>0.1) suggested the occurrence of a
prolonged dust event in Shanghai. By using the $PM_{2.5}/PM_{10}$ mass ratio of 0.4 as a
threshold (Fan et al., 2021), the period from October 29 to November 2 was defined as
the dust period in this study. The remaining days, including October 25 to October 28
and November 3 to November 6, were defined as the non-dust period. Throughout the
entire dust period, the mean concentrations of $PM_{2.5}$ and $PM_{10}$ reached 53.3 ±
20.5μg/m$^3$ and 172.4 ± 70.2μg/m$^3$, respectively, yielding a low $PM_{2.5}/PM_{10}$ ratio of 0.34
± 0.15. As a comparison, $PM_{2.5}$ and $PM_{10}$ during the non-dust period was 38.9μg/m$^3$
and 49.8μg/m$^3$, respectively, exhibiting a relatively high $PM_{2.5}/PM_{10}$ ratio of 0.62 ± 0.20.

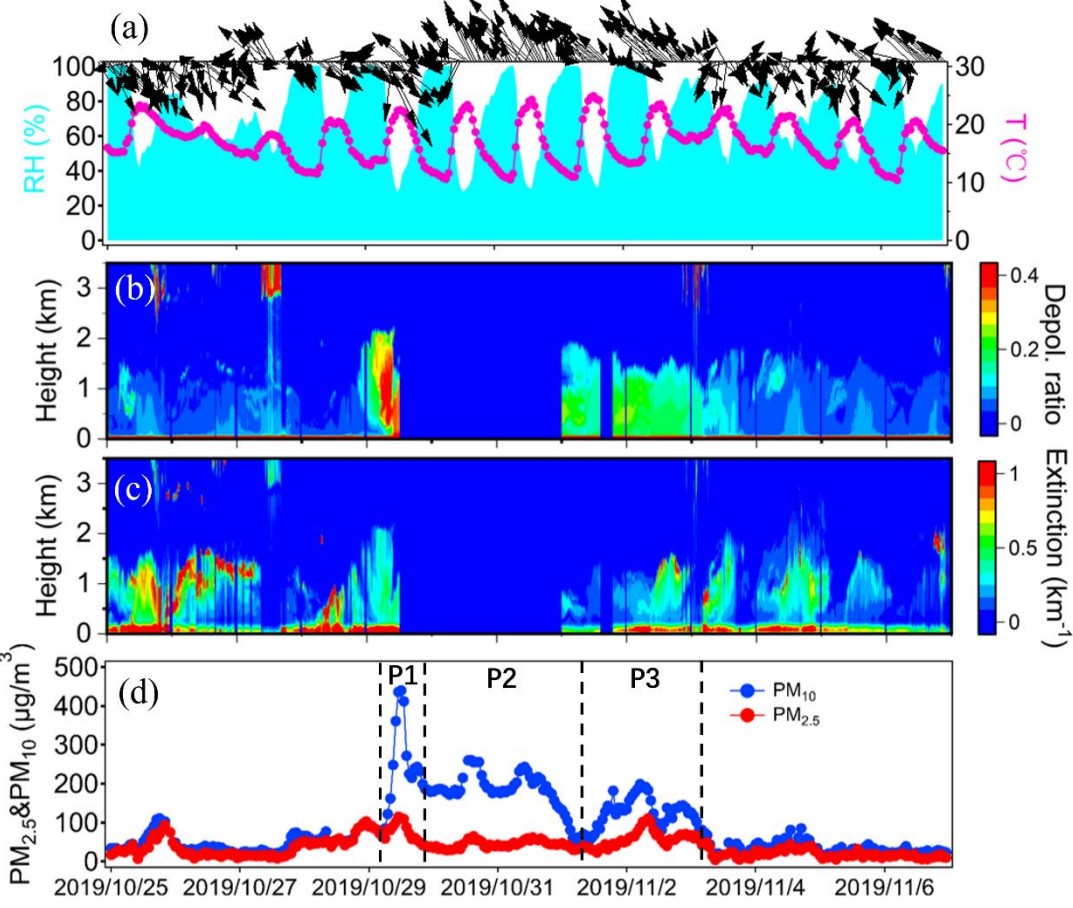


Figure 1. Time series of (a) relative humidity, temperature, wind vectors, (b) aerosol
depolarization ratio, (c) aerosol extinction coefficient, (d) mass concentrations of $PM_{2.5}$
and $PM_{10}$ during the study period. Three dust stages, i.e., P1, P2, and P3 are also marked.
The missing aerosol lidar data were due to instrument malfunction.

The occurrences of dust are typically accompanied by low relative humidity and
strong winds due to the passage of cold fronts ( Huang et al., 2010; Wang et al., 2013;
Wang et al., 2018). However, in this study, relative humidity was exceptionally high
with the mean value of 71±26%. It showed strong diurnal variation, reaching its
minimum in the daytime and even close to 100% in the nighttime (Figure 1a).
Additionally, wind speed was low of 0.54±0.59m/s with a maximum of 2.6m/s. This
stagnant synoptic condition led to elevated concentrations of main gaseous pollutants
such as $O_3$, $SO_2$, and $NO_2$, with mean values of $86.0\pm47.8\mu g/m^3$, $11.8\pm3.4\mu g/m^3$, and
$63.3\pm27.9\mu g/m^3$, respectively, even higher than those during the non-dust period.
We further divided the dust event into three stages based on the temporal
characteristics of $PM_{10}$ and the transport patterns of air masses. As shown in Figure 1d,
$PM_{10}$ quickly climbed from 4:00 on October 29 and reached a maximum of $436\mu g/m^3$
after 8 hours. The air masses primarily originated from the semi-arid regions of
northwest China (Figure 2d), which was consistent with both the near surface wind
observation (Figure 1a) and wind lidar observation (Figure 2a). The wind profiles
showed prevailing northwest winds from the surface up to around 2km before noon on
October 29, indicating the presence of a strong synoptic system. Afterwards, $PM_{10}$
quickly decreased to $199\ \mu g/m^3$ at 20:00, October 29 within 8 hours. This was primarily
attributed to the shift of wind directions. As shown in Figure 2a, while winds above
700m continued to blow from the northwest, near- surface winds had shifted from the
southeast. Due to Shanghai's coastal location adjacent to the East China Sea, the
relatively clean southeasterlies diluted the local air pollutants, thereby explaining the
quick decline in PM10 concentrations. This initial dust episode occurring from 4:00 -
13:00 on October 29 was defined as Stage P1.
Despite the persistent southeasterly winds, the dust event did not come to a
complete halt. Even under these prevailing winds, hourly $PM_{10}$ concentrations
remained above $150\ \mu g/m^3$ until November 1, gradually decreasing to $65\ \mu g/m^3$ at 03:00,
November 1 (Figure 1d). Compared to P1, wind speed during this stage was as low as
$0.4\pm0.5m/s$, while RH was moderately high of $70\pm26\%$. Although the daytime RH
stayed low between 30% and 50%, it frequently soared above 90% at nighttime. Figure
2e shows that although the air masses originated from the Gobi Desert, they also
traversed considerable coastal regions. The wind profiles further indicated that while
northwest winds prevailed at altitudes higher than 500m, east and northeast winds were
dominant below 500m (Figure 2b). Consequently, the relatively high RH during this
period can be attributed to the mixing of dust plumes with coastal sea breezes. This dust
episode from 14:00, October 29 to 3:00 on November 1 was designated as Stage P2.
Following P2, $PM_{10}$ and $PM_{2.5}$ rose again and peaked at 5:00 and 9:00 on
November 2 with the hourly concentration of 199 and $117\mu g/m^3$, respectively. Different
from P1 and P2, the air masses during this stage originated from the Shandong
Peninsula and the northern region of Jiangsu province, and then migrated over the
Yellow Sea and the East China Sea (Figure 2f). Typically, dust plumes tend to travel
eastward, impacting the western Pacific region and even distant oceanic regions (Wang
et al., 2018; Nagashima et al., 2016). However, in this case, the air masses evidently
deviated and pushed the dust back towards the mainland. The wind profiles on
November 2 revealed that winds within the detected altitude range predominantly
originated from the eastern and southeastern oceanic regions (Figure 2c). This probably
indicated the mixing between dust plumes and humid oceanic air masses was quite
sufficient, which was also reflected by the highest average RH of $76 \pm 24\%$ among the
three stages of the dust event. Moreover, the concentrations of $O_3$ and $NO_2$ at this stage
were higher than those of P1 and P2, potentially promoting the formation of secondary
aerosol components and will be discussed later. This rarely observed dust backflow
transport episode from 4:00 on November 1 to 23:00 on November 2 was designated
as Stage P3.

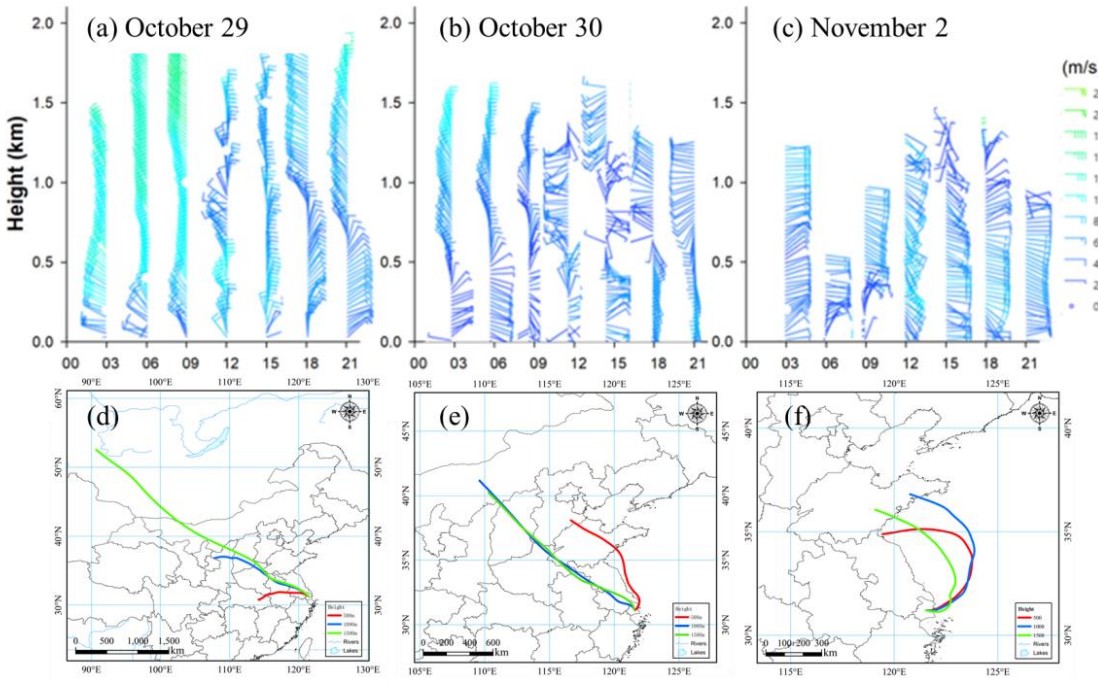

Figure 2. Wind profiles observed by a wind profiler radar on (a) October 29, (b) October 30, and (c) November 2. 48-hour backward trajectories simulated at the sampling site starting from (d) 4:00 AM, October 29, (e) 9:00 AM, October 30, and (f) 13:00 PM, November 2. The red, blue, and green trajectories represented starting altitudes of 100, 500, and 1500m, respectively.

## 3.2. Comparisons of aerosol chemical compositions among the three dust stages

Figure 3a shows the time-series of hourly aerosol chemical components, including SNA ($NO_3^-$, $SO_4^{2-}$, and $NH_4^+$), OM, EC, and mineral aerosols in $PM_{2.5}$. During P1, the mean concentration of SNA was $49.9 \pm 31.6$ μg/m$^3$. The mineral aerosols reached $16.4 \pm 14.6$ μg/m$^3$, accounting for 19% in $PM_{2.5}$. The contribution of

OM to PM$_{2.5}$ was almost identical to that of mineral aerosols (Figure 3b).

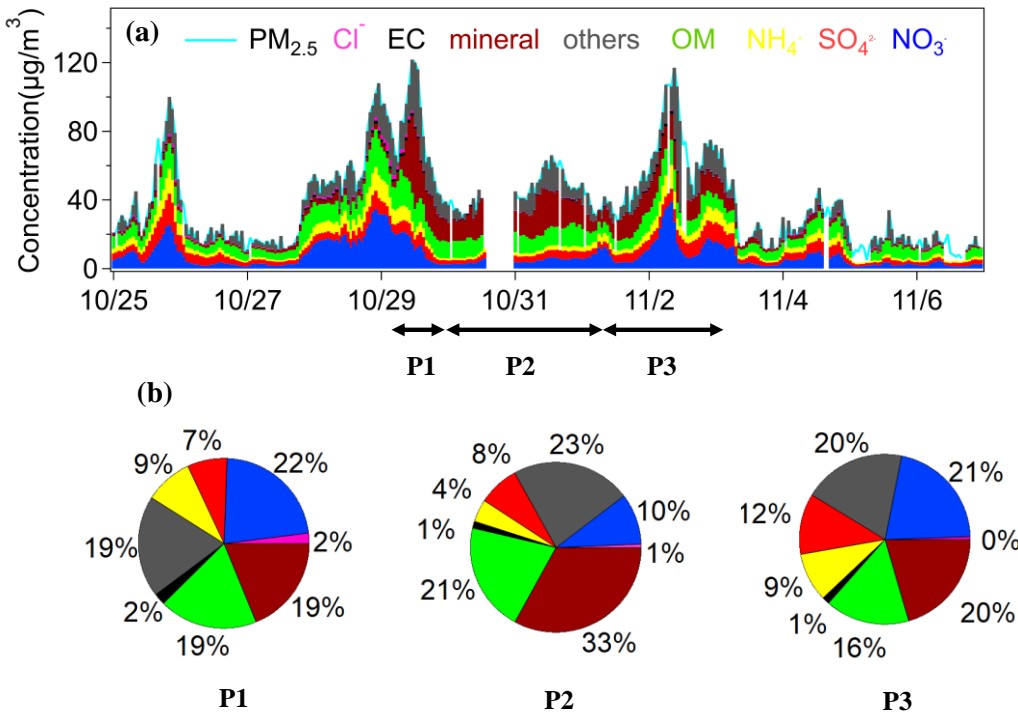

Figure 3. (a) Times-series of major chemical components in PM$_{2.5}$ during the study
period. (b) The mean proportion of major chemical components in PM$_{2.5}$ during the
three dust stages.

During P2, mineral aerosols increased to 23.4±54.1μg/m$^3$ and accounted for 33%

in PM$_{2.5}$, representing the highest among all three stages (Figure 3b). Due to the
continuous dilution effect of dust on local anthropogenic pollutants, the concentrations
and proportions of SNA in PM$_{2.5}$ were the lowest during this stage. For instance, NO$_3^-$
only accounted for 10% in PM$_{2.5}$, indicating a suppression of nitrate formation to some
extent. The levels of OM didn't exhibit obvious changes and averaged 10.1±2.1μg/m$^3$,
accounting for 21% in PM$_{2.5}$.

During P3, mineral aerosols averaged 11.9±2.7μg/m$^3$, ranking the lowest among

all three stages. The proportion of mineral aerosols in PM$_{2.5}$ decreased to 20%,
suggesting the dust backflow from the ocean was less enriched in mineral components.
Compared to P2, SNA showed significant increases and much stronger diurnal
variations during P3. $SO_4^{2-}$, $NO_3^-$, and $NH_4^+$ averaged $6.7 \pm 2.4$, $12.4 \pm 8.9$, and $5.4 \pm$
$2.7\mu g/m^3$, respectively. As shown in Figure 3b, the contribution of nitrate to $PM_{2.5}$
increased to 21%, while that of sulfate rose to 12%, the highest among all three stages.
The concentration of OM ($9.3\pm3.2\mu g/m^3$) and its proportion (16%) during P3 were
lower than the other two stages, likely due to the unconventional dust backflow
transport pathway.

**3.3. Unconventional features of the dust episodes**
**3.3.1. Unusually enhanced $O_3$ during dust**
Figure 4 shows the hourly near surface ozone concentrations and vertical profiles
of ozone during the study period. Interestingly, a few high $O_3$ peaks occurred during
the dust event (Figure 4a). $O_3$ averaged $92.8 \pm 52.8\mu g/m^3$ during the dust, about 50%
higher than the non-dust days. Among the three dust stages, $O_3$ substantially increased
from $35.9 \pm 36.4\mu g/m^3$ during P1 to $80.7 \pm 41.2\mu g/m^3$ during P2, and further rose to
$104.0 \pm 48.7\mu g/m^3$ during P3. The low $O_3$ during P1 can be attributed to the cleansing
effect of the strong dust associated with the cold front, which was consistent with
previous studies that reported reduced oxidant concentrations during intense dust
events (Benas et al., 2013). Regarding the relatively high $O_3$ during P2 and P3, several
causes may contribute to this phenomenon. Firstly, the mean wind speed was low of
0.4 and 0.6 m/s during P2 and P3, respectively. One numerical study conducted
during the similar period suggested that the low wind speed caused reduction of
boundary layer height and the warming of the lower atmosphere, thus accelerating the
ozone formation by ~1 ppbv/h (Wang et al., 2020). Consequently, this weak synoptic
system was favorable for the accumulation of ozone. Secondly, since the dust plume
travelled mostly over the coastal and oceanic areas, a portion of $O_3$ could be
transported from the high ozone oceanic areas (Wang et al., 2022b). Thirdly, the
ozone lidar also detected high $O_3$ stripes during P2 and P3. As shown in Figure 4b, the
high $O_3$ profiles extended from the surface to around 1km and the profile structure
was similar to that of aerosol depolarization ratio. The subsidence of dust particles
likely contributed to downward transport of high-altitude $O_3$, thereby influencing the
elevated $O_3$ near the ground (Yang et al., 2022).

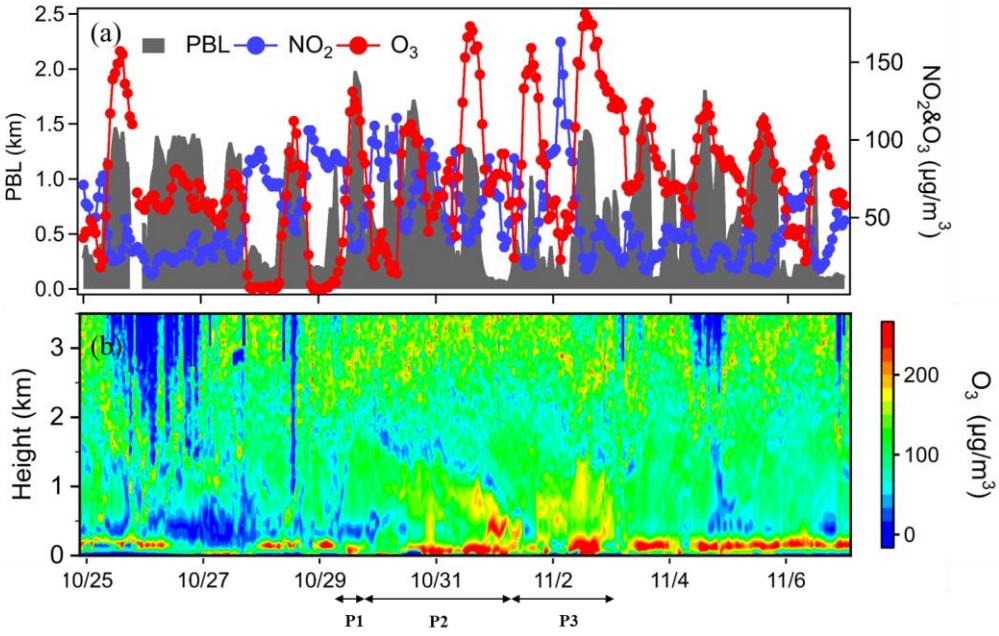


Figure 4. (a) Time-series of near surface $O_3$, $NO_2$ and planetary boundary layer height
(b) Vertical profiles of ozone observed by the ozone Lidar.

**3.3.2. Dust backflows during P3**
The dust during P3 was diagnosed as a backflow transport pathway from the
mainland to Shanghai through the Yellow Sea and the East China Sea, as determined
by the backward trajectory analysis (Figure 2f). This unconventional dust transport
route, termed "dust backflow", was consistent with a similar occurrence in 2014 when
dust from northern China detoured over Shanghai (Wang et al., 2018). In this section,
we have provided further evidences of the dust backflow from various aspects.
Figure 2f illustrates that the dust drifted away from the Shandong Peninsula, thus
we selected two coastal sites in Shandong province for supplementary analysis. Figure
S2 compares the time-series of hourly air pollutants at Qingdao, Lianyungang, and
Shanghai. At Qingdao and Lianyungang, high $PM_{10}$ concentrations were observed
during October 30 – 31, indicating the invasion of dust in these regions. After about
two days, $PM_{10}$ peaked in Shanghai on early November 2. This temporal consistency
aligned with the simulation duration of the backward trajectories, which lasted around
48 hours (Figure 2f). In Figure 9, it can be observed that in the upstream dust regions
(i.e., Qingdao and Lianyungang), $PM_{10}$ varied negatively with $NO_2$ and CO (the
highlighted period in the figure). While in Shanghai, positive correlations between
$PM_{10}$ and both $NO_2$ ($R^2$=0.32) and CO ($R^2$=0.55) indicated that the dust during P3
served as a carrier for gaseous pollutants rather than acting a diluter.


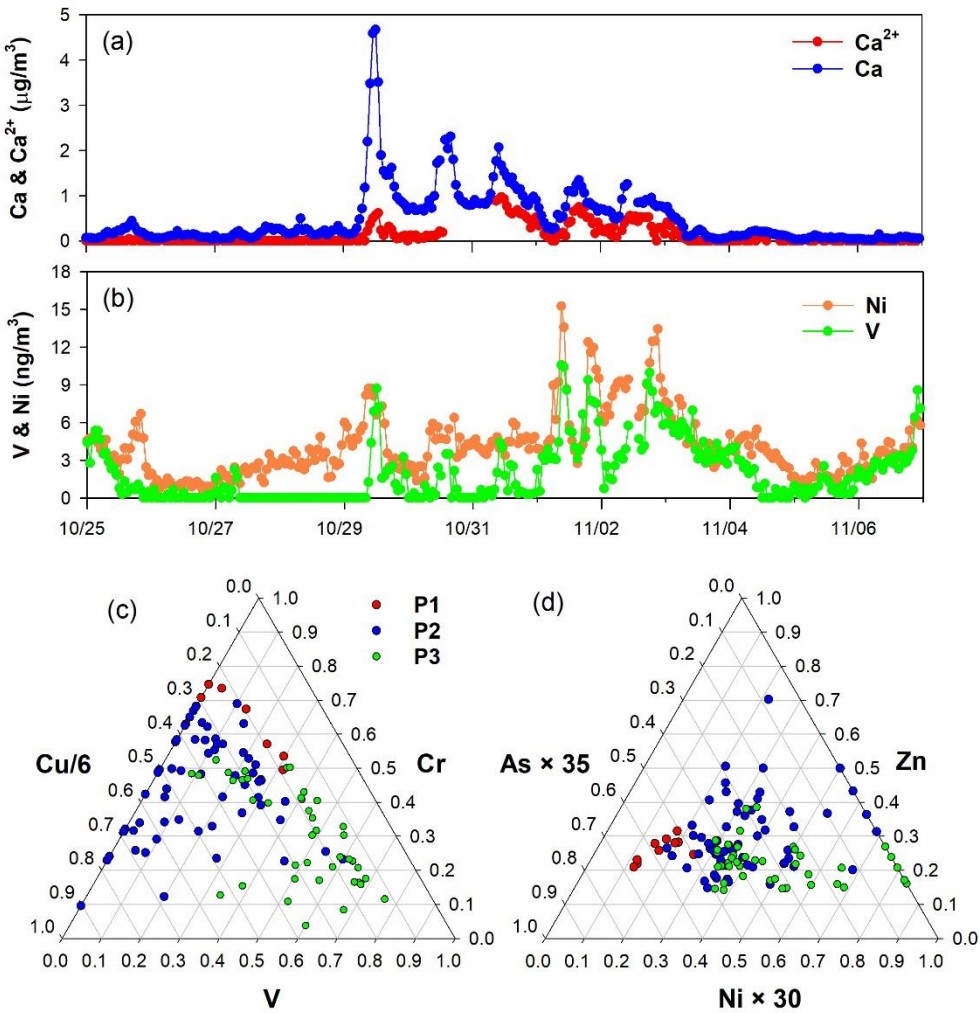


Figure 5. Time-series of (a) Ca, Ca$^{2+}$, (b) V, and Ni during the study period. (c)

Cu-Cr-V ternary diagram and (d) As-Zn-Ni ternary diagram among the P1 – P3 dust

episodes. Due to substantial concentration differences of various elements, some

elements are artificially changed to make most scatters appear around the centroid.

Additional evidence of dust backflows was provided from the perspective of

aerosol chemical tracers. Figure 5a plots the time-series of Ca and Ca$^{2+}$, representing

the total calcium and the soluble part of calcium, respectively. It was observed that Ca

and Ca$^{2+}$ didn't exhibit a proportional variation trend, which should be related to the

solubility of calcium during different dust stages. During P1, the mean concentration of
Ca reached the highest of $1.63 \pm 1.53\mu g/m^3$ while $Ca^{2+}$ was the lowest of $0.21 \pm$
$0.20\mu g/m^3$, thus resulting in the lowest $Ca^{2+}/Ca$ ratio of $0.10 \pm 0.08$. As discussed in
Section 3.1, dust during P1 was the strongest and thus it contained higher fractions of
minerals, primarily in the form of insoluble metal oxides. The average concentrations
of $Ca^{2+}$ and Ca during P2 were $0.33 \pm 0.28\mu g/m^3$ and $1.11 \pm 0.46\mu g/m^3$, respectively,
resulting in the higher $Ca^{2+}/Ca$ ratio of $0.27 \pm 0.20$. As a comparison, the average
concentrations of $Ca^{2+}$ and Ca during P3 reached $0.34 \pm 0.20\mu g/m^3$ and $0.78 \pm$
$0.27\mu g/m^3$, respectively, yielding the highest $Ca^{2+}/Ca$ ratio of $0.38 \pm 0.19$. The
significantly higher solubility of calcium during P3 should be directly related to the
prolonged presence of dust plumes over the open ocean. The abundant water vapor over
the ocean could accelerate the dissolution of the insoluble components in particles
during the mixing between continental dust and oceanic air masses. Additionally, the
backflow transport pathway facilitated the entrainment of sea salts and contributed to
the increase of soluble calcium.
Figure 5b provides additional insights by displaying the time-series of V and Ni,
which are typical tracers of oil combustions (Becagli et al., 2012). They varied
significantly during the study period, and the mass concentrations of V and Ni increased
4 and 1.8 times during P3 compared to P2, respectively.    Consistently, the enrichment
factors of Ni and V displayed higher values during P3 than P1 and P2 (Figure S3). The
trends are substantiated in the ternary diagrams, which are commonly applied to
illustrate the relative abundances of three components and infer the source variations
(Bozlaker et al., 2019; Cwiertny et al., 2008; Laskin et al., 2005). As shown in the Cu-
Cr-V ternary diagram (Figure 5c), the dust samples during P1 were positioned away
from the V-apex. As a comparison, the dust samples during P2 exhibited greater
scattering, manifesting enhanced anthropogenic contributions, e.g., from chrome
plating industries (Hammond et al., 2008). Compared to P2, the dust samples during P3
moved toward the V-apex, indicating a higher contribution from oil combustions
(Becagli et al., 2012). A similar pattern was observed in the As-Zn-Ni ternary diagram
(Figure 5d). The majority of dust samples during P2 spanned across the diagram,
reflecting contributions from mixed anthropogenic sources. Reciprocally, P3 was closer
to the Ni-apex. These lines of evidences collectively confirmed that the dust had mixed
with pollutants from marine vessel emissions over one of the busiest international
shipping trade routes (Fan et al., 2016) and was subsequently transported back to
Shanghai.

**3.4. Formation of secondary aerosols during the dust long-range transport**
**3.4.1. Comparison of typical chemical tracers**

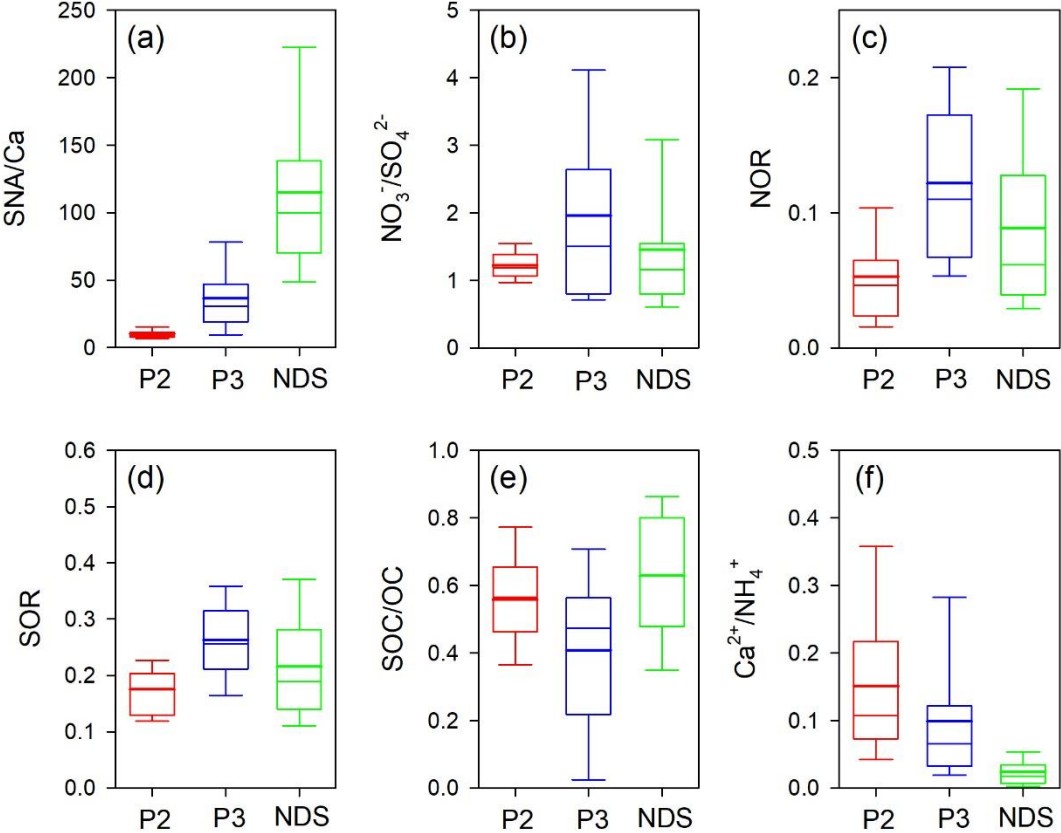


Figure 6. Box plots of (a) SNA/Ca, (b) $NO_3^-/SO_4^{2-}$, (c) NOR, (d) SOR, (e) SOC/OC,

and (f) $Ca^{2+}/NH_4^+$ during P2, P3, and NDS, respectively.

To delve deeper into the formation characteristics of secondary aerosols in

different stages, a variety of chemical tracers was investigated. The P1 dust stage was
excluded from statistical analysis due to its short duration and limited data availability.
Figure 6a shows the comparison of SNA/Ca ratios during P2, P3 and non-dust days
(NDS). The SNA/Ca ratio can be used to assess the relative changes between secondary
production and primary dust emission by eliminating the impact of meteorological
conditions among different periods (Zheng et al., 2015). Compared to the two dust
episodes, the SNA/Ca ratio is significantly higher during NDS. This can be attributed
to the much lower concentrations of mineral aerosols during NDS, thus resulting in the
higher SNA relative to Ca. In terms of comparing P2 and P3, the average SNA/Ca ratio
during P3 was 3 times that of P2, indicating that the formation of secondary inorganic
aerosols was more prominent during the dust backflow. Regarding the $NO_3^-/SO_4^{2-}$ ratios
(Figure 6b), they were close between NDS and P2, with $NO_3^-$ slightly exceeding $SO_4^{2-}$.
The range of $NO_3^-/SO_4^{2-}$ was the largest during P3 with a mean value of around 2,
suggesting that the dust backflow was more conducive to the accumulation of nitrate.
The nitrogen oxidation ratio ($NOR = NO_3^-/(NO_3^- + NO_2)$) and the sulfur oxidation ratio
($SOR = SO_4^{2-}/(SO_4^{2-} + SO_2)$) were further used to gauge the extent of nitrate and sulfate
formation, both showing trends of P3>NDS>P2 (Figure 6c & 6d). It should be noted
that NOR and SOR cannot be used to realistically characterize the extent of nitrogen
and sulfur oxidation during transport-dominated pollution cases, as upstream aging
aerosols can significantly increase the above ratios (Ji et al., 2018). In the following
discussion, we will focus on the formation mechanism of SNA during different dust
stages.
The results of SOC/OC ratios differed from the above analysis that SOC/OC was
lower during P3 than during P2 and NDS (Figure 6e), suggesting that the formation of
secondary organic aerosols was not favored via the dust backflow. This may be due to
its maritime transport pathway as the emission intensity of volatile organic compounds
from the ocean is much lower than that from land sources. Consequently, the lacking of
organic aerosol precursors could be the main cause for the lower SOC/OC ratios during
P3. Finally, the $Ca^{2+}/NH_4^+$ ratio was employed to assess the relative contributions of
alkaline chemical components (Figure 6f). As expected, this ratio during the two dust
stages was much higher than that of NDS, indicating the important contribution of dust
to alkaline metal ions. The $Ca^{2+}/NH_4^+$ ratio was higher during P3 (0.15) than during P2
(0.10), which aligned with the findings presented in Section 3.2.

### 3.4.2. Distinct formation processes of secondary aerosols between P2 and P3


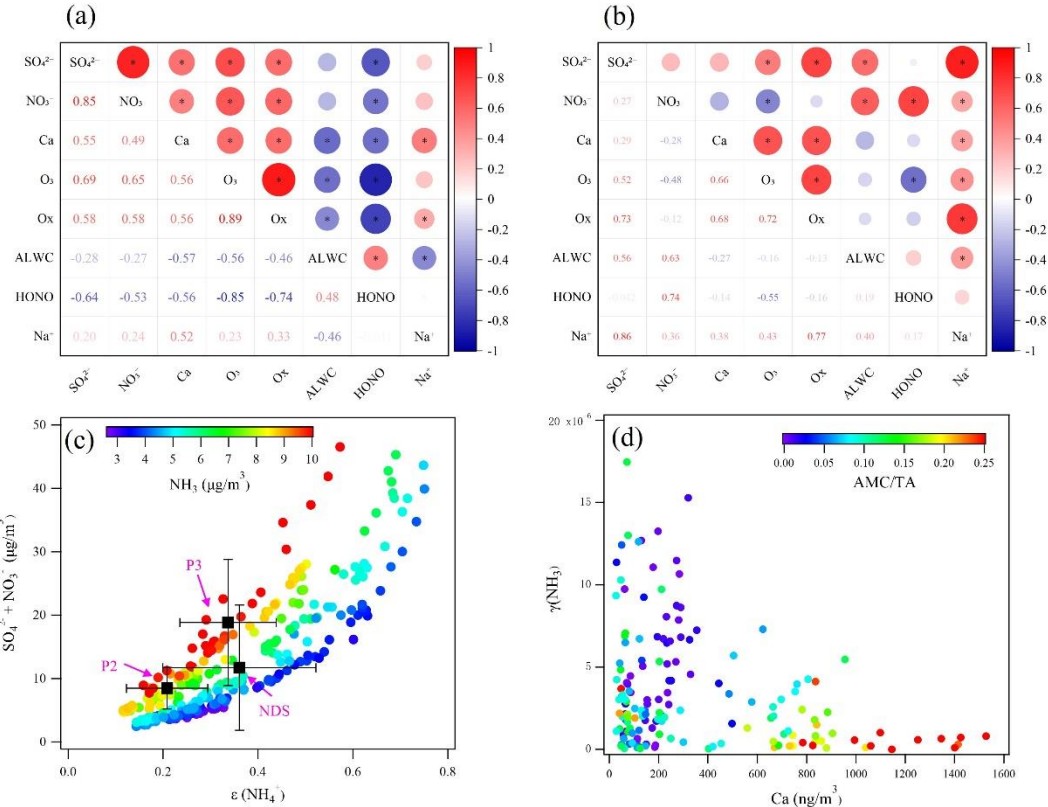


Figure 7. Correlation heatmaps during (a) P2 and (b) P3. The size of dot corresponds to the value of the correlation coefficient. The black star inside the dot means the correlation is significant ($p<0.05$). (c) The mass concentrations of $SO_4^{2-}$ and $NO_3^-$ as a function of $\varepsilon(NH_4^+)$, with dots colored by the concentration of $NH_3$. The mean states of P2, P3, and NDS are added. Error bars denote one standard deviations. (d) The uptake coefficient of $NH_3$ ($\gamma_{NH3}$) on particles as a function of Ca, with dots colored by the AMC/TA molar ratios. AMC and TA denote the total molar concentrations of $Na^+$, $K^+$, $Mg^{2+}$, and $Ca^{2+}$ and the total molar concentrations of anions, respectively.


In this section, we further analyze the formation mechanism and key influencing
factors of secondary components during P2 and P3. Figure 7a & 7b compare the
correlation heatmaps of $SO_4^{2-}$ and $NO_3^-$ with various parameters. During P2, both $SO_4^{2-}$
and $NO_3^-$ displayed the most significant correlations with $O_3$ and Ox ($O_3+NO_2$), while
even negatively correlated with ALWC. In regard of the obvious ozone enhancement
phenomenon as discussed in Section 3.3.1, the photochemistry pathway for the
secondary aerosol formation (e.g., $S(IV) + O_3$ (aq) $\rightarrow$ $S(VI)$) should overwhelm over
the aqueous phase pathways, e.g., oxidation by $H_2O_2$, catalysis by tracer metals, and
oxidation by $NO_2$. In addition, $SO_4^{2-}$ and $NO_3^-$ also showed moderate correlations with
elemental Ca, suggesting that dust acted as a carrier to transport these salts, which can
be derived from background minerals in dust (Wu et al., 2022) and dust heterogeneous
reactions during the transport (Huang et al., 2010).
As for P3, it showed a distinctly different correlation heatmap from P2. While
$SO_4^{2-}$ still demonstrates a correlation with $O_3$, the relationship between $NO_3^-$ and $O_3$ (as
well as Ox) disappeared. On the contrary, both $SO_4^{2-}$ and $NO_3^-$ show significant
correlations with ALWC. During P3, the average RH reached 76%, providing favorable
conditions for liquid-phase reactions. Furthermore, by relating $NO_3^-$ and the
multiplication of ALWC and $NO_2$, the correlation coefficient ($R^2 = 0.41$) was further
improved (Figure S4a). Similar results were observed by relating $NO_3^-$ to the
multiplication of ALWC and $NO_2*O_3*NO_2$ (a proxy of $N_2O_5$, (Huang et al., 2021))
(Figure S4b), confirming the dominant reaction pathway of nitrogen oxides to nitrate
via the aqueous phase reactions. As a result, $NO_3^-$ was also strongly correlated with
HONO (Figure S4c), typically deriving from the heterogeneous reactions of $NO_2$ on
the surface of moist particles (Alicke et al. (2002).

In addition, unlike P2, both $SO_4^{2-}$ and $NO_3^-$ showed moderate to significant

correlations with $Na^+$, a tracer of sea salts (Figure 7b). Since neither $SO_4^{2-}$ nor $NO_3^-$
correlated with Ca, it can be inferred that sea salts played a more important role in the
transport of air pollutant during the dust backflow over the ocean. To assess whether
dust or sea salts participated in the heterogeneous reactions of secondary aerosol during
P3, the ISORROPIA II model was run with different scenarios. Figure S5 shows the
model performance for $SO_4^{2-}$, $NO_3^-$, $NH_4^+$, and $NH_3$ based on the
$SO_4^{2-}-NO_3^--NH_4^+-Cl^--NH_3-HCl-HNO_3$ system. After adding $Ca^{2+}$ into this
thermodynamic equilibrium system, the correlations between the simulations vs
observations for all four species were lowered with different extents (Figure S6). If $Na^+$
was added into the thermodynamic equilibrium system. the model performance was
slightly improved (Figure S7). This corroborated that the heterogeneous reactions on
dust were very limited while sea salts were intensively involved in the formation of
secondary inorganic aerosols during the dust backflow.

To further explore the influencing factors affecting the formation of secondary

inorganic aerosols, we examined the role of $NH_3$ in different stages, representing by the
relationship between the gas-particle partitioning of ammonia ($\varepsilon(NH_4^+)$, defined as the
ratio between particle phase ammonia ($NH_4^+$) and total ammonia ($NH_x = NH_3+NH_4^+$))
and the total acids ($SO_4^{2-} + NO_3^-$). As shown in Figure 7c, it is obvious that the total
acids strongly co-varied with $\varepsilon(NH_4^+)$. Higher $\varepsilon(NH_4^+)$ resulted in higher
concentrations of secondary aerosols. Moreover, under similar $\varepsilon(NH_4^+)$ conditions,
higher $NH_3$ promoted stronger formation of secondary aerosols. Thus, both $NH_3$ and
$\varepsilon(NH_4^+)$ collectively determined the aerosol formation potential. The mean states of
P2, P3, and NDS are compared in Figure 7c. P2 had the lowest $\varepsilon(NH_4^+)$ with the mean
value of 0.21, despite the relatively high concentrations of NH3 during this period (7.9
$\pm$ 1.0 $\mu$ g/m$^3$). The relatively low gas-particle partitioning of ammonia limited the
neutralization of the acidic components. In contrast, NH3 during P3 was the highest
during the study period (9.8 $\pm$ 1.8 $\mu$g/m$^3$), and $\varepsilon(NH_4^+)$ (0.34) was only slightly lower
than that during NDS, thus effectively fostering the formation of secondary inorganic
aerosols.

To explain this phenomenon, the uptake coefficient of NH3 ($\gamma_{NH3}$) on particles,

which is one of the important parameters affecting the gas-particle partitioning of
ammonia, was calculated. Figure 7d shows the decreasing trend of $\gamma_{NH3}$ with the
increase of dust intensity (using Ca as an indicator). This coincided with a multi-year
observational study in Beijing and Shijiazhuang, where $\gamma_{NH3}$ obviously increased due
to significant decline in alkali earth metal contents from the dust emission sources
during 2018 – 2020 (Liu et al., 2022). Thus, this partially explained why $\varepsilon(NH_4^+)$ was
relatively low during P2, which was ascribed to the reduced uptake capacity of NH3 on
particles.

The ion balance calculation indicated that the total anions and cations are in ideal

equilibrium (Figure S8, regression slope = 0.99, R$^2$ = 0.99), indicating that both NH4$^+$
and alkali metal cations (including Na$^+$, K$^+$, Mg$^{2+}$, and Ca$^{2+}$) contributed to the
neutralization of acids to varying degrees. The ratio of alkali metal cations/total anions
(AMC/TA) was used to color the data points in Figure 7d, showing an opposite trend
between AMC/TA and $\gamma_{NH3}$. During P2, the mean value of AMC/TA reached 21%,
implying that the neutralization of acids by NH3 had been significantly suppressed, thus
explaining `the decrease in the NH3 uptake coefficient at high dust intensity. In contrast,
the AMC/TA ratio decreased to 11% during P3, indicating a reduced competition
between $NH_3$ and the alkali dust components. Finally, we also compared the aerosol pH
at different stages, which was 3.2, 3.0, and 2.8 during P2, P3, and NDS, respectively.
The relatively high aerosol acidity at P3 and NDS favored the uptake of alkaline gases
(Liu et al., 2022), which also contributed to the higher ($\gamma_{NH3}$) at these two stages.

**3.5. Estimation of transported and secondarily formed particles during P3**
As discussed in previous sections, the aerosols observed during P3 could originate
from both aged aerosols transported via the dust backflows and secondary formation.
In this section, we aimed to estimate the contribution of transport and secondary
formation to the main aerosol species, respectively, based on the simultaneous
measurements at the Pudong site and the Lianyungang site. As discussed in Section
3.4.1, Lianyungang acted as an upstream region where dust drifted away from the
mainland. The duration of dust observed at Lianyungang was approximately from 5:00
on October 30 to 16:00 on October 31, about 46 hours ahead of the dust invasion
observed at Pudong (Figure S2).
To assess the extents of transported air pollutants, black carbon (BC) was used as
a reference aerosol component. As shown in Figure S9, one BC pollution episode on
October 30 at Lianyungang was observed. Correspondingly, another BC pollution
episode emerged at Pudong on November 2after about 46 hours. Since the air mass
trajectory from Lianyungang to Pudong predominantly traversed over the ocean, and
considering that BC has no secondary sources, it can be reasonably assumed that the
differences of BC concentrations between these two sites were ascribed to the removal
processes of particles.
To determine the removal fractions of aerosols during dust transport, we first
defined the average concentrations of various aerosol components during the preceding
five hours of the dust at Pudong as their background concentrations. Then, a coefficient
$k$ was derived to calculate the removal fractions of aerosols during the dust transport as
below.

$$k = \frac{AV_{LYG,BC} - (AV_{PD,BC} - BKG_{PD,BC})}{AV_{LYG,BC}}$$

(3)

$AV_{LYG,BC}$ and $AV_{PD,BC}$ represent the average concentration of BC at Lianyungang
and Pudong during their respective dust period. $BKG_{PD,BC}$ represents the background
concentration of BC at Pudong. Assuming that other aerosol species were removed with
a similar efficiency as BC, the amounts of transported aerosol species from
Lianyungang to Pudong can be estimated as below.
$TP_{PD,i} = AV_{LYG,i} \times (1 - k)$      (4)
$TP_{PD,i}$ represents the transported amounts of aerosol species $i$. $1-k$ represents the
transport fraction of aerosols. Then, the secondarily formed aerosol species $i$ at Pudong
can be calculated as below.
$SF_{PD,i} = AV_{PD,i} - BKG_{PD,i} - TP_{PD,I}$      (5)
Figure 8 shows the results of the transported and the secondarily formed aerosol
species during P3. It was calculated that the secondarily formed and transported $NO_3^-$
averaged 6.8μg/m$^3$ and 4.7μg/m$^3$, accounting for about 45% and 31% of its total mass
concentration, respectively. This was consistent with the earlier analysis that a
considerable portion of nitrate was formed through the aqueous phase secondary
formation. In contrast, it was calculated that the transported $SO_4^{2-}$ accounted for about

42% of its total mass concentration, while the secondarily formed $SO_4^{2-}$ was almost negligible. This was also consistent with the phenomenon that $SO_4^{2-}$ correlated significantly with $Na^+$ (Figure 6b). As for $NH_4^+$, it exhibited a similar apportionment as $NO_3^-$, with the secondarily formed and transported $NH_4^+$ accounting for about 35% and 28% of its total mass concentration, respectively. Compared to $NO_3^-$ and $NH_4^+$, OM was more dominated by transport (57%) while its secondary formation only accounted for about 13%. It should be noted that the simple method devised in this study may have inherent uncertainties. Considering the prolonged duration of the dust event, it is possible that certain dust particles lingered over the open ocean. Consequently, the contributions attributed to aerosol transport should be considered as a conservative estimate or lower bound, rather than an exhaustive assessment.

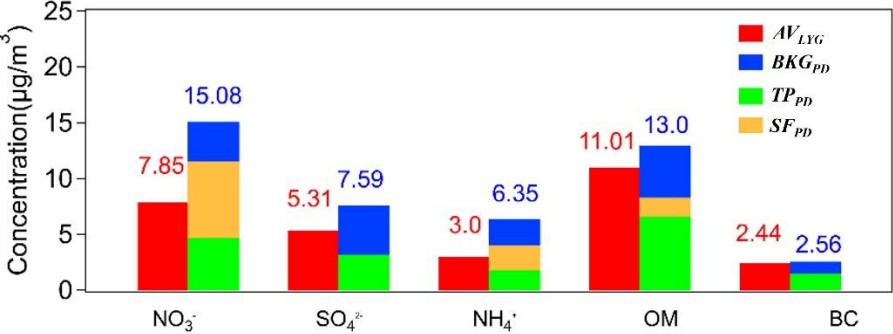

Figure 8. The apportioned concentrations of the major aerosol species during P3.

## 4. Conclusion

During October 29 to November 2, 2019, a long-lasting dust event was observed in Shanghai based on a synergy measurement of near surface air pollutants, aerosol lidar,

wind profiling lidar, and air masses trajectory modeling. Different from most dust
events, this dust event was characterized of exceptionally high relative humidity, low
wind speed, and relatively high concentrations of gaseous pollutants. The dust event
was divided into three stages from P1 – P3. P1 was a short dust episode due to the
strong cold front. P2 was a dust episode when RH was moderately high of $70 \pm 26\%$
and the southeasterlies prevailed with partial air masses from coastal regions. P3 was a
rarely observed dust backflow transport episode. The air masses originated from the
Shandong Peninsula and the northern region of Jiangsu province, and then migrated
over the Yellow Sea and the East China Sea. RH reached the highest of $76 \pm 24\%$ among
the three stages of the dust event.
During P2, abnormally high $O_3$ concentrations were observed, which could be due
to the weak synoptic system as well as down drafted high-altitude $O_3$ along with the
subsidence of dust particles Sulfate and nitrate moderately correlated with $O_3$ while had
almost no correlation with ALWC, indicating that the formation of secondary aerosols
during P2 should be mainly promoted via the gas-phase oxidations. During P3, a special
phenomenon of dust backflow was observed and confirmed by various evidences. The
highest $Ca^{2+}/Ca$ ratio was observed due to that the lingerer of dust plumes over the open
ocean. Moreover, contributions of V and Ni significantly increased, indicating the
mixing between dust and marine vessel emissions. Different from P2, nitrate
significantly correlated with ALWC but not with $O_3$, indicating its aqueous-phase
reaction pathway. Also, sulfate and nitrate exhibited moderate to strong correlations
with $Na^+$, suggesting sea salts as a medium for the heterogeneous reactions.
By analyzing various chemical tracers, the formation extent of SNA was found
much stronger during P3 than during P2. Both $NH_3$ and $\varepsilon(NH_4^+)$

666 ($NH_4^+$/($NH_3$+$NH_4^+$)) determined the concentrations of SNA. To explain the relatively

667 high $\varepsilon(NH_4^+)$ values during P3, the uptake coefficient of $NH_3$ ($\gamma_{NH3}$) on particles is

668 calculated. $\gamma_{NH3}$ negatively varied with the intensity of dust, which were attributed to

669 two factors. Higher contributions of alkali metal components suppressed the

670 neutralization capacity of $NH_3$ on acids, thereby lowering $\gamma_{NH3}$ during P2. Also,

671 relatively high aerosol pH during P2 didn't facilitate the uptake of $NH_3$ and the

672 subsequent aerosol formation.

673  Based on a simplified method, the amounts of transported and secondarily formed

674 particles during P3 were quantified. It was calculated that about 45% and 31% of $NO_3^-$

675 was contributed by secondary formation and transport, respectively. In contrast, the

676 transported $SO_4^{2-}$ accounted for about 42% of its total mass concentration while the rest

677 was from its background concentration with negligible secondary formation. OM was

678 dominated by transport (57%) while its secondary formation only accounted for about

679 13%.

680

681 **Data Availability Statement**

682 All data used in this study can be requested upon the corresponding author

683 (huangkan@fudan.edu.cn).

684

685 **Author contributions**

686  KH, QF, and YD designed this study. JH, FY, YL, and JC performed data

687 collection. DL and KH performed data analysis and wrote the paper. All have

688 commented on and reviewed the paper.


**Competing interests**
The authors declare that they have no conflict of interest.

**Acknowledgments**
This work was financially supported by the National Key R&D Plan programs
(2023YFE0102500) and National Science Foundation of China (42175119).

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
