# Peer review of "Secondary aerosol formation under a special dust transport event: impacts from unusually enhanced ozone and dust backflows over the ocean"

_EGUsphere, 2023_

## Referee Comment (RC2)

Desert dust aerosol has great impacts on atmospheric chemistry. Although there are numerous field measurements in this field, the work by Lu et al. is very interesting as the dust event examined was very unique, characterized by low wind speed, high RH, higher concentration of reactive trace gases, and dust backflow. This work can be published after the following comments are addressed.

**Major comments:**

I have to admit that I am not familiar with field observation work, and therefore the editor may invite colleagues with field observation expertise to assess this manuscript. In my opinion, the phenomena Lu et al. observed and tried to interpret is very interesting, but the discussion can still be improved. For example, the two most important sections (3.3 and 3.4) are quite descriptive, and conclusions they drew are not very well supported in the current version. Therefore, further data analysis is encouraged, and the authors may think about if they could deeper insights into dust chemistry from the unique event they observed.

**Minor comments:**

Line 53: The authors may consider citing a recent review paper (Tang et al., 2017) which discussed impacts of dust on tropospheric chemistry.

Line 71: Guo et al. (2019) investigated hygroscopicity of a number of Ca and Mg salts, and suggested that atmospheric aging could greatly enhance hygroscopicity of mineral dust.

Line 379-382, line 395-398: Sulfate and nitrate formation is quite complicated in the troposphere. These conclusions currently drawn by the authors are merely based on correlations, and further discussion is needed.

Line 436-451: Sea spay aerosols also contain soluble $Ca^{2+}$. The authors may want to subtract the contribution of sea spray aerosol on soluble $Ca^{2+}$ using the $Ca^{2+}/Na^+$ ratio in sea water.

Line 493-494: This sentence sound strange. I am not sure if sulfate can be aged in the atmosphere, as sulfate cannot be further oxidized. In addition, the co-variation of sulfate with $Na^+$ may indicate important contribution of sea spray aerosol (which is a major source of $Na^+$) to sulfate.

**References:**

Tang, M. J., Huang, X., Lu, K. D., Ge, M. F., Li, Y. J., Cheng, P., Zhu, T., Ding, A. J., Zhang, Y. H., Gligorovski, S., Song, W., Ding, X., Bi, X. H., and Wang, X. M.: Heterogeneous reactions of mineral dust aerosol: implications for tropospheric oxidation capacity, Atmos. Chem. Phys., 17, 11727-11777, 2017.

---

## Author Comment (AC1)

**Response to Reviewer #1's Comments**

In this manuscript, the authors focused on the atmospheric physiochemical characterizations of a transported dust event that passed through the city of Shanghai and then circled back again. However, it was written in poor English and lacking for scientific questions/gaps, which made it just like a data analysis note of various instruments of a super monitoring site based on three stages of dust event. In general, this manuscript has no innovations and valuable contributions for the science community, and it should be reorganized and discussed with other authors. As it presented in low quality, and I cannot accept it for the publication in present status, and it should be rejected.

We thank for the reviewer's comments and suggestions on this manuscript. Based on the specific comments, we have substantially reorganized the whole manuscript and made numerous changes. We have responded to all the comments point-by-point and made corresponding changes in the manuscript as highlighted in the track change mode. Please check the detailed responses to all the comments as below, and we hope to receive further comments based on this version.

Major questions:

1.  The title of this manuscript is obscure and unmatched to the research contents.
The original title is not precise. In the revision, the title is changed as "Secondary aerosol formation under a special dust transport event: impacts from unusually enhanced ozone and dust backflows over the ocean".

2.  The Introduction part had lengthy writing and lacking for logics among paragraphs. Each paragraph also lacks the topic sentence. The mainstream references are lacking in this part. As mentioned above, the key defect is lacking of research gaps and scientific questions.
Thanks for suggestion. We have removed all the redundant references and writings. We

have also updated more related studies. Some paragraphs have been reorganized, and we have made the goal of this study more clearly written in the last paragraph in the introduction.

Since there are numerous changes in the introduction section, please check the changes in **Line 59 - 191**.

3.    The math formulas should be provided in the methodology part, such as Lines 292-293 and Lines 535-550.

Thanks for the suggestion. The formulas in the original Lines 292-293 have been moved to the methodology part. As for the method in Lines 535-550, it is based on the discussion of dust backflows, thus we prefer to state it in the main text.

In the revision, we added a section about the uptake coefficient of $NH_3$ ($\gamma_{NH3}$) on particles in the methodology part. $\gamma_{NH3}$ is used to deeply analyze the formation of secondary aerosol formation. Please check the changes in **Lines 263 - 277**.

4.    The authors claimed that they develop "a simple method of relating the upstream and receptor simultaneous measurements", however, this method lacks for the basic scientific verifications, such as comparisons with other source apportionment methods (PMF, CMB, etc.).

Thanks for the suggestion and we do agree that this method should be compared with other methods. We have conducted PMF modeling and seven sources are identified as shown in the figure below. It can be seen sulfate and nitrate have very low loadings in the identified dust factor or sea salt factor. This means dust have contributed very little to the secondary aerosols based on the PMF results. This is expected as although PMF is a powerful source apportionment tool, it is not designed to apportion the contributions from secondary formation and long-range transport.

In this study, dust transport is frequent and the formation of sulfate and nitrate evidently occurred during the transport. However, it is clear that both the local formation and long-range transported sulfate and nitrate have been grouped into Factor 1 and Factor 2 of the PMF modeling. Thus, we need to develop other methods for the separation of

transported aerosols and secondary formed aerosols.

[Figure]

5.    Too many figures in this manuscript, and the authors can only use the linear analysis in them, and this reflects that this manuscript is written by a student and lacking for the basic guidance.

Thanks for the suggestion and we quite agree with reviewer that linear analysis is overused in the original manuscript. In the revision, we have removed all figures from Figure 6 to Figure 12. The original writings in Section 3.3 & 3.4 have been almost deleted. We have fully reorganized the analysis and the major changes are summarized below.

**3.3.2. Dust backflows during P3**

In this section, Ternary diagrams of typical metals are added to support the analysis of dust backflows. Please check the detailed writing in **Lines 541 – 581.**

[Figure]

**3.4.1. Comparison of typical chemical tracers**

In this section, we analyzed a set of chemical tracers to gain insights into the formation characteristics of secondary aerosols in different stages, please check the full writings in **Lines 583 – 621.**

[Figure]

**3.4.2. Distinct formation processes of secondary aerosols between P2 and P3**

In this section, we further analyze the formation mechanism and key influencing factors of secondary components during P2 and P3. Major changes are made as below. (1) The correlation heatmaps of $SO_4^{2-}$ and $NO_3^-$ with various parameters are compared between P2 and P3. (2) The role of the gas-particle partitioning of $NH_3$ in the formation of secondary aerosols is analyzed. (3) The uptake coefficient of $NH_3$ on particles is applied to explain the aerosol formation potentials in different stages. Please check the full writings in **Lines 623 – 705.**

[Figure]

The whole paper needs to be rewritten in scientific language and vocabularies, and it is necessary that a native English speaker need to help the authors to revised it word by word.

This manuscript is now fully rewritten after polishing the language and vocabularies. It is revised by a native English speaker. Please check the track changed version to see all the changes.

**Response to Reviewer #2's Comments**

Desert dust aerosol has great impacts on atmospheric chemistry. Although there are numerous field measurements in this field, the work by Lu et al. is very interesting as the dust event examined was very unique, characterized by low wind speed, high RH, higher concentration of reactive trace gases, and dust backflow. This work can be published after the following comments are addressed.

We thank for the reviewer's positive comments and helpful suggestions on this manuscript. Based on the specific comments, we have responded to all the comments point-by-point and made corresponding changes in the manuscript as highlighted in the track change mode. Please check the detailed responses to all the comments as below.

Major comments: I have to admit that I am not familiar with field observation work, and therefore the editor may invite colleagues with field observation expertise to assess this manuscript. In my opinion, the phenomena Lu et al. observed and tried to interpret is very interesting, but the discussion can still be improved. For example, the two most important sections (3.3 and 3.4) are quite descriptive, and conclusions they drew are not very well supported in the current version. Therefore, further data analysis is encouraged, and the authors may think about if they could deeper insights into dust chemistry from the unique event they observed.

Thanks for the comments. We quite agree with the reviewer that the analysis in the original version was quite descriptive and more deeper data analysis is needed. In the revision, we have removed all the descriptive analysis and related discussions in Section 3.3 and 3.4. Instead, we added two new sections in the revised manuscript.

**3.4.1. Comparison of typical chemical tracers**

In this section, we analyzed a set of chemical tracers to gain insights into the formation characteristics of secondary aerosols in different stages, please check the full writings

[Figure]

**3.4.2. Distinct formation processes of secondary aerosols between P2 and P3**

In this section, we further analyze the formation mechanism and key influencing factors of secondary components during P2 and P3. Major changes are made as below. (1) The correlation heatmaps of $SO_4^{2-}$ and $NO_3^-$ with various parameters are compared between P2 and P3. (2) The role of the gas-particle partitioning of $NH_3$ in the formation of secondary aerosols is analyzed. (3) The uptake coefficient of $NH_3$ on particles is applied to explain the aerosol formation potentials in different stages. Please check the full writings in **Lines 623 – 705.**

[Figure]

Minor comments:

Line 53: The authors may consider citing a recent review paper (Tang et al., 2017) which discussed impacts of dust on tropospheric chemistry.

Thanks for proving the information. The reference is added as below.

Tang et al. (2017) conducted a comprehensive review on the effect of dust heterogeneous reactions on the tropospheric oxidation capacity. They proposed that high RH (e.g., > 80%) and a wider range of temperature should be considered in the laboratory studies of heterogeneous reactions of mineral dust. Also, more comprehensive kinetic models should be developed to understand the complex multiphase reactions.

Line 71: Guo et al. (2019) investigated hygroscopicity of a number of Ca and Mg salts, and suggested that atmospheric aging could greatly enhance hygroscopicity of mineral dust.

Thanks for proving the information. The reference is added as below.

Guo et al. (2019) investigated the dependence of deliquescence relative humidities on temperature and hygroscopic properties of eight Ca- and Mg-containing salts. It was found that the hygroscopic growths of some salts were significant at 90 % RH, implying that favorable environmental conditions could greatly enhance the hygroscopicity of mineral dust.

References:

Tang, M. J., Huang, X., Lu, K. D., Ge, M. F., Li, Y. J., Cheng, P., Zhu, T., Ding, A. J., Zhang, Y. H., Gligorovski, S., Song, W., Ding, X., Bi, X. H., and Wang, X. M.: Heterogeneous reactions of mineral dust aerosol: implications for tropospheric oxidation capacity, Atmos. Chem. Phys., 17, 11727-11777, 2017.

Guo, L., Gu, W., Peng, C., Wang, W., Li, Y. J., Zong, T., Tang, Y., Wu, Z., Lin, Q., Ge, M., Zhang, G., Hu, M., Bi, X., Wang, X., and Tang, M.: A comprehensive study of hygroscopic properties of calcium- and magnesium-containing salts: implication for hygroscopicity of mineral dust and sea salt aerosols, Atmos. Chem. Phys., 19, 2115–2133, https://doi.org/10.5194/acp-19-2115-2019, 2019.

Line 379-382, line 395-398: Sulfate and nitrate formation is quite complicated in the troposphere. These conclusions currently drawn by the authors are merely based on correlations, and further discussion is needed.

Thanks for the comments and we quite agree. As responded above, we have removed almost all the original writings and added deeper analysis in Section 3.4. Please check the detailed writings in **Lines 583 – 705.**

Line 436-451: Sea spay aerosols also contain soluble Ca2+. The authors may want to subtract the contribution of sea spray aerosol on soluble Ca2+ using the Ca2+/Na+ ratio in sea water.

Thanks for the comment. It is a good idea that the contribution of sea spray aerosol on soluble $Ca^{2+}$ can be quantified using the $Ca^{2+}/Na^+$ ratio in sea water. However, considering that during the dust period, $Na^+$ could be derived from both dust and sea salts. Thereby, the contribution of sea spray aerosol on soluble $Ca^{2+}$ could be overestimated by using $(Ca^{2+}/Na^+)_{seawater} * Na^+_{aerosol}$. In the revision, we have added the sentence "Additionally, the backflow transport pathway facilitated the entrainment of sea salts and contributed to the increase of soluble calcium." in **Lines 558 - 559.**

Line 493-494: This sentence sound strange. I am not sure if sulfate can be aged in the atmosphere, as sulfate cannot be further oxidized. In addition, the co-variation of sulfate with Na+ may indicate important contribution of sea spray aerosol (which is a major source of Na+ ) to sulfate.

Thanks for pointing out this incorrect statement. We didn't mean sulfate can be aged but meant to say part of sulfate was formed in the upstream regions and transported with the sea spray aerosols. In the revision, we have re-written this paragraph in Lines 655 - 667.

In addition, unlike P2, both $SO_4^{2-}$ and $NO_3^-$ showed moderate to significant correlations with $Na^+$. Since neither $SO_4^{2-}$ nor $NO_3^-$ correlated with Ca, it can be inferred that sea salts played a more important role in the transport of air pollutant during the dust backflow over the ocean. To assess whether dust or sea salts participated in the heterogeneous reactions of secondary aerosol during P3, the ISORROPIA II model was run with different scenarios. Figure S5 shows the model performance for $SO_4^{2-}$, $NO_3^-$, $NH_4^+$, and $NH_3$ based on the $SO_4^{2-}-NO_3^--NH_4^+-Cl^--NH_3-HCl-HNO_3$ system. After adding $Ca^{2+}$ into this thermodynamic equilibrium system, the correlations between the simulations vs observations for all four species were lowered with different extents (Figure S6). If $Na^+$ was added into the thermodynamic equilibrium system. the model performance was slightly improved (Figure S7). This corroborated that the heterogeneous reactions on dust were very limited while sea salts were intensively involved in the formation of secondary inorganic aerosols during the dust backflow.

References: Tang, M. J., Huang, X., Lu, K. D., Ge, M. F., Li, Y. J., Cheng, P., Zhu, T., Ding, A. J., Zhang, Y. H., Gligorovski, S., Song, W., Ding, X., Bi, X. H., and Wang, X. M.: Heterogeneous reactions of mineral dust aerosol: implications for tropospheric oxidation capacity, Atmos. Chem. Phys., 17, 11727-11777, 2017.

---

## Referee Report (RR1)

Manuscript Number: egusphere-2023-127
Title: Secondary aerosol formation under a special dust transport event: impacts from unusually enhanced ozone and dust backflows over the ocean

General comments

This study reports a dust event with high relative humidity and low wind speeds occurred in Shanghai during the period of October 29-November 2, 2019. The dust event was divided into three obvious stages, of which the first stage was a dust invasion stage, the second stage was a dust development period, and the third stage was a dust backflow period. Meanwhile, chemical characteristics of aerosols in the three stages were investigated, and a simplified method was deployed to identify and estimate the amounts of major aerosol species from transport and secondary formation. The study method is reasonable, data is reliable, and conclusion is credible. But there existed a lot of aspects to be revised and improved in the manuscript. Written language and logical relationship need to be improved. I suggest to consider the paper for publication after a major revision.

Specific comments

1. In abstract, L28, the phrase "by with high concentrations of particulate matters but relatively short duration" should be changed to "by high concentrations of particulate matters but relatively short duration".

2. In introduction, L121, the sentence "In contrast, this study aims to depict an atypical dust event was observed in Shanghai, a coastal mega-city in Eastern China." is error.

3. In section 2.2, all the online instruments used in this study should be normally calibrated so as to guarantee data quality, therefore, normal calibrations of these monitoring instruments should be added in the section.

4. In sections 2.3 and 2.4, the models used in this study should be detailed.

5. In section 2.5, the sentence "Initially, the quasi-first-order reaction rate constant for heterogeneous conversion from NH3 to NH4+ ($k_{het}$, $s^{-1}$) is calculated according to (Liu et al., 2022)." is incomplete. In addition, the two formulas should be numbered. All the formulas in this study should be numbered in turn.

6. In section 2 methodology, sampling duration should be supplemented.

7. In section 3.1, L216-218, the sentence "From October 25 to 28, the mean wind speed remained relatively low of 0.9±0.72m/s with a peak value of 3.1m/s, and predominantly blowing from the northwest." is suggested to be changed to "From October 25 to 28, the mean wind speed was 0.9±0.72m/s with a peak value of 3.1m/s, remaining relatively

low, and predominantly blowing from the northwest."

8. L218, the sentence "The mean concentration of PM2.5 and PM10 was 34.7 and 44.2 µg/m3, respectively." should be revised to "The mean concentrations of PM2.5 and PM10 were 34.7 and 44.2 µg/m3, respectively."

9. L214-228, in this paragraph dust and non-dust periods should be identified, but authors did not give related discussion.

10. In Figure 1, P1, P2 and P3 should be put in Figure 1d and separated with vertical lines.

11. What did aerosol depolarization ratio was used to explain? Please explain correlation between the depolarization ratio values and the dust event and impacts of relative humidity on the depolarization ratio.

12. L237-244, "In this study………non-dust period", please supply specific start and end time.

13. Huang et al., 2010a and Huang et al., 2010b were the same reference. Please check the reference.

14. L337-342, in "Firstly, …... ~1 ppbv/h (Wang et al., 2020).", related explanation is lack of logic. Please think it over and revise the explanation. In the text, there are many similar logical problems need to be further checked and revised.

15. L345, Figure 5b here should be Figure 4b.

16. In correlation heatmaps, please explain the meaning of dot size.

17. L494, how to understand the sentence "both SO42- and NO3- showed moderate to significant correlations with Na+."

18. The mean states of P2, P3 and NDS illustrated in Figure 7c were unclear, hope to better present these states.

19. In section 3.5, in the formula $TP_{PD,i} = AV_{LYG,i} \times (1-k)$, please explain the meaning of "$1-k$".

20. The conclusions need to be further condensed.

---

## Referee Report (RR2)

Manuscript Number: egusphere-2023-127
Title: Secondary aerosol formation under a special dust transport event: impacts from unusually enhanced ozone and dust backflows over the ocean

The authors have revised the related contents according to the comments and answered all the related questions. I suggest to accept the paper. The authors should pay attention to giving the correct line numbers of the revised contents in the "Response to the Comments".

---

## Author Response (AR2)

**Response to Anonymous Referee #3's Comments**

General comments

This study reports a dust event with high relative humidity and low wind speeds occurred in Shanghai during the period of October 29-November 2, 2019. The dust event was divided into three obvious stages, of which the first stage was a dust invasion stage, the second stage was a dust development period, and the third stage was a dust backflow period. Meanwhile, chemical characteristics of aerosols in the three stages were investigated, and a simplified method was deployed to identify and estimate the amounts of major aerosol species from transport and secondary formation. The study method is reasonable, data is reliable, and conclusion is credible. But there existed a lot of aspects to be revised and improved in the manuscript. Written language and logical relationship need to be improved. I suggest to consider the paper for publication after a major revision.

We sincerely thank for the reviewer's positive comments and helpful suggestions on this manuscript. Based on the specific comments, we have responded to all the comments point-by-point and made corresponding changes in the manuscript as highlighted in red color. We feel the revisions based on the reviewer's comments have greatly improved the quality of this manuscript. Please check the detailed responses to all the comments as below.

Specific comments

1. In abstract, L28, the phrase "by with high concentrations of particulate matters but relatively short duration" should be changed to "by high concentrations of particulate matters but relatively short duration".

Thanks for pointing out this typo. The word "with" is now deleted in the revision.

2. In introduction, L121, the sentence "In contrast, this study aims to depict an atypical

dust event was observed in Shanghai, a coastal mega-city in Eastern China." is error.

This is indeed a grammatical error. The sentence is now revised as "In contrast, this study aims to depict an atypical dust event that was observed in Shanghai, a coastal mega-city in Eastern China.".

3. In section 2.2, all the online instruments used in this study should be normally calibrated so as to guarantee data quality, therefore, normal calibrations of these monitoring instruments should be added in the section.

Thanks for the suggestion. For all the online instruments equipped at the Pudong supersite, they are routinely maintained by professional technicians. The online data are checked each day via an online integrated information system. Thus, the data quality can be assured. In Line 182 of the revised manuscript, the sentence "All instruments are routinely maintained and calibrated to ensure the quality of data." is added.

4. In sections 2.3 and 2.4, the models used in this study should be detailed.

In Line 187-205, Sections 2.3 is revised as below.

The ISORROPIA II model is subject to the principle of minimizing the Gibbs energy of the multi-phase aerosol system, leading to a computationally intensive optimization problem (Song et al., 2018). The model can predict the physical state and compositions of atmospheric inorganic species ($NH_4^+$, $Na^+$, $K^+$, $Mg^{2+}$, $Ca^{2+}$, $SO_4^{2-}$, $NO_3^-$ and $Cl^-$) with their gas- and particle-phase concentrations and meteorological parameters (relative humidity and temperature) as model inputs. The model includes two modes, i.e., reverse and forward mode. The reverse mode calculates the equilibrium partitioning based on aerosol-phase concentrations only, while the latter uses both aerosol-phase and gas-phase concentrations as inputs. Moreover, particles can be assumed as "metastable" with liquid-phase but no solid participating while "stable" with the liquid and solid phases or both. The ISORROPIA running in the forward mode at the metastable state was applied in this study. Aerosol pH was calculated based on the equilibrium particle hydronium ion concentration and aerosol liquid water content (ALWC) obtained from model results. The performances and advantages of

ISORROPIA over the usage of other thermodynamic equilibrium codes has been assessed in numerous studies (Nenes et al., 1998; West et al., 1999; Ansari and Pandis, 1999; Yu et al., 2005).

In Line 209 – 221, Sections 2.4 is revised as below.

The HYSPLIT (Hybrid Single-Particle Lagrangian Integrated Trajectory) was used to compute the backward trajectories of the air parcels during the dust events. It is a widely used model that computes dispersion following the particle or puff. The advection of a particle or puff is computed from the average of the three-dimensional velocity vectors for the initial position and the first-guess position (Draxler and Hess, 1998). Turbulent velocity components are expressed as a function of the velocity variance, a statistical quantity derived from the meteorological data, and the Lagrangian time scale. The calculation of air mass trajectories can be used to depict the airflow patterns for interpreting the transport of air pollutants over various spatial and temporal ranges (Stein et al., 2015). In this study, the HYSPLIT model was driven by meteorological data outputs from the Global Data Assimilation System (GDAS) (Su et al., 2015), which is available at ftp://arlftp.arlhq.noaa.gov/pub/archives/gdas1. Air mass trajectories were launched at different heights from the ground and a total duration of 48 hours simulation was conducted.

References:

Draxler, R. R., and G. D. Hess, 1998: An overview of the HYSPLIT_4 modeling system for trajectories, dispersion, and deposition. Aust. Meteor. Mag., 47, 295–308.

Stein, A. F., Draxler, R. R., Rolph, G. D., Stunder, B. J. B., Cohen, M. D., Ngan, F., NOAA's HYSPLIT Atmospheric Transport and Dispersion Modeling System, Bulletin of the American Meteorological Society, 2015, 2059–2077.

5. In section 2.5, the sentence "Initially, the quasi-first-order reaction rate constant for heterogeneous conversion from NH3 to NH4+ (khet, s-1) is calculated according to (Liu et al., 2022)." is incomplete. In addition, the two formulas should be numbered.

All the formulas in this study should be numbered in turn.

This sentence is revised as "Initially, the quasi-first-order reaction rate constant for heterogeneous conversion from $NH_3$ to $NH_4^+$ ($k_{het}$, $s^{-1}$) is calculated by Eq. (1) (Liu et al., 2022)." in Line 228.

All the formulas in this study have been numbered in the revision.

6. In section 2 methodology, sampling duration should be supplemented.

In the revision, we have added the sampling duration for the applied instruments as below.

Line 162-163: Samples were collected for 40 min and then analyzed in the following 20 min.

Line 171-172: These parameters were measured at the temporal resolution of 5min.

Line 174: Meteorological parameters (ambient temperature, relative humidity, wind speed, and wind direction) were obtained by a Vaisala Weather transmitter (WXT520) at the temporal resolution of 1min.

Line 175-181: The height of planetary boundary layer (PBL) was retrieved from a ceilometer (CL31, Vaisala) at the temporal resolution of 30 min. Vertical profiles of aerosol optical properties were obtained by an aerosol lidar (AGJ, AIOFM) at the temporal resolution of 30 min and vertical resolution of 7.5 m, respectively. Vertical profiles of ozone were obtained by an ozone lidar (LIDAR-G-2000, WUXIZHONGKE) at the temporal resolution of 15 min and vertical resolution of 7.5 m, respectively.

7. In section 3.1, L216-218, the sentence "From October 25 to 28, the mean wind speed remained relatively low of 0.9±0.72m/s with a peak value of 3.1m/s, and predominantly blowing from the northwest." is suggested to be changed to "From October 25 to 28, the mean wind speed was 0.9±0.72m/s with a peak value of 3.1m/s, remaining relatively low, and predominantly blowing from the northwest."

Thanks for the suggestion. It is revised in Line 244-245 as suggested.

8. L218, the sentence "The mean concentration of PM2.5 and PM10 was 34.7 and 44.2

μg/m3, respectively." should be revised to "The mean concentrations of PM2.5 and PM10 were 34.7 and 44.2 μg/m3, respectively."

It is corrected in Line 247 as suggested.

9. L214-228, in this paragraph dust and non-dust periods should be identified, but authors did not give related discussion.

In this paragraph, the definitions of dust and non-dust periods have been added in Line 255-263.

By using the $PM_{2.5}/PM_{10}$ mass ratio of 0.4 as a threshold (Fan et al., 2021), the period from October 29 to November 2 was defined as the dust period in this study. The remaining days, including October 25 to October 28 and November 3 to November 6, were defined as the non-dust period. Throughout the entire dust period, the mean concentrations of $PM_{2.5}$ and $PM_{10}$ reached $53.3 \pm 20.5 \mu g/m^3$ and $172.4 \pm 70.2 \mu g/m^3$, respectively, yielding a low $PM_{2.5}/PM_{10}$ ratio of $0.34 \pm 0.15$. As a comparison, $PM_{2.5}$ and $PM_{10}$ during the non-dust period was $38.9 \mu g/m^3$ and $49.8 \mu g/m^3$, respectively, exhibiting a relatively high $PM_{2.5}/PM_{10}$ ratio of $0.62 \pm 0.20$.

Reference:

Fan, H., Zhao, C., Yang, Y., and Yang, X.: Spatio-Temporal Variations of the PM2.5/PM10 Ratios and Its Application to Air Pollution Type Classification in China, Frontiers in Environmental Science, 2021, https://doi.org/10.3389/fenvs.2021.692440.

10. In Figure 1, P1, P2 and P3 should be put in Figure 1d and separated with vertical lines.

As suggested, Figure 1 is revised as below.

[Figure]

11. What did aerosol depolarization ratio was used to explain? Please explain correlation between the depolarization ratio values and the dust event and impacts of relative humidity on the depolarization ratio.

The depolarization ratio, a measure of the irregularity of the scatterer shape, is the most important property of dust measured by lidar systems (Shimizu et al., 2017). The high depolarization ratio of aerosol was due to the nonsphericity (irregular shapes) and relatively large size of particles (Mcneil and Carswell, 1975). If the depolarization ratio of the region is less than 10% it is identified as spherical aerosol, and if it exceeds 10% it is identified as mineral dust (Shimizu et al., 2004).

In general, if relative humidity is high, particles can absorb more water and thus become more spheric, thus lowering the depolarization ratio, and vice versa.

In the revision (Line 251 - 254), we have made changes to describe more clearly about

the depolarization ratio.

In general, if the particle depolarization ratio exceeds 10%, the aerosol is identified as mineral dust (Shimizu et al., 2004) due to the nonsphericity (irregular shapes) and relatively large size of particles (Mcneil and Carswell, 1975).

References:

Shimizu, A., Nishizawa, T., Jin, Y., Kim, S. W., Wang, Z., Batdorj, D., and Sugimoto, N.: Evolution of a lidar network for tropo-spheric aerosol detection in East Asia, Opt. Eng., 56, 031219, https://doi.org/10.1117/1.oe.56.3.031219, 2017.

Shimizu, A., N. Sugimoto, I. Matsui, K. Arao, I. Uno, T. Murayama, N. Kagawa, K. Aoki, A. Uchiyama, and A. Yamazaki (2004), Continuous observations of Asian dust and other aerosols by polarization lidars in China and Japan during ACE-Asia, J. Geophys. Res.,109, D19S17, doi:10.1029/ 2002JD003253.

Mcneil, W. R. and Carswell, A. I.: Lidar Polarization Studies of Troposphere, Appl. Opt., 14, 2158–2168, 1975.

12. L237-244, "In this study………non-dust period", please supply specific start and end time.

In Line 257-259, the non-dust period is defined as "October 25 to October 28 and November 3 to November 6".

13. Huang et al., 2010a and Huang et al., 2010b were the same reference. Please check the reference.

Thanks for pointing out this mistake. It is corrected in the text and reference section.

14. L337-342, in "Firstly, …... ~1 ppbv/h (Wang et al., 2020).", related explanation is lack of logic. Please think it over and revise the explanation. In the text, there are many similar logical problems need to be further checked and revised.

This paragraph is revised in Line 375-380 as below.

Firstly, the mean wind speed was low of 0.4 and 0.6 m/s during P2 and P3, respectively.

One numerical study conducted during the similar period suggested that the low wind speed caused reduction of boundary layer height and the warming of the lower atmosphere, thus accelerating the ozone formation by ~1 ppbv/h (Wang et al., 2020). Consequently, this weak synoptic system was favorable for the accumulation of ozone.

15. L345, Figure 5b here should be Figure 4b.

Thanks for pointing out this typo. It is corrected in the revision.

16. In correlation heatmaps, please explain the meaning of dot size.

The size of dot means the value of the correlation coefficient, corresponding to the numbers in the heatmaps. Also, the black star inside the dot means the correlation is significant (p<0.05). In the caption of Figure 7, the meaning of dots is added in Line 503-507.

17. L494, how to understand the sentence "both SO42- and NO3- showed moderate to significant correlations with Na+."

This sentence is based on the correlation analysis as shown in Figure 7b. We meant to say that by using $Na^+$ as the tracer of sea salts, the moderate to significant correlations between $SO_4^{2-}$/$NO_3^-$ and $Na^+$ indicated the contribution of sea salts to the secondary aerosol formation during the dust backflow over the ocean.

In the revision (Line 539), we have stated more clearly as below.

In addition, unlike P2, both $SO_4^{2-}$ and $NO_3^-$ showed moderate to significant correlations with $Na^+$, a tracer of sea salts (Figure 7b).

18. The mean states of P2, P3 and NDS illustrated in Figure 7c were unclear, hope to better present these states.

In the revised Figure 7c, the mean states of P2, P3 and NDS have been illustrated more clearly as below. In addition, the meaning of P2, P3 and NDS is added in the caption of Figure 7.

[Figure]

19. In section 3.5, in the formula TPPD,i = AVLYG,i × (1-k), please explain the meaning of "1-k ".

k *is* defined as the removal coefficient of aerosols during the dust transport. Thus, 1-k means the transport fraction of aerosols. In the revision, the meaning of "1-k" is explained in Line 623-624.

20. The conclusions need to be further condensed.

Thanks for the suggestion. Some redundant writings are removed. Please check Line 651-705 for the changes.

**Response to Anonymous Referee #4's Comments**

The quality of the revised manuscript is significantly improved compared to the original version. The authors adequately addressed the questions raised by the reviewers. Overall, I support the publication of the manuscript upon addressing the following questions.

We sincerely thank again for the reviewer's further assessment on this manuscript. Based on the specific comments, we have responded to all the comments point-by-point and made corresponding changes in the manuscript as highlighted in red color. We feel the revisions based on the reviewer's comments have greatly improved the quality of this manuscript. Please check the detailed responses to all the comments as below.

(1) For the analysis of sulfate formation, I'm still not convinced. In several places (i.e., Lines 473-475 and Lines 623-625), the authors suggested sulfate showed correlation with O3 or O3+NO2, but not with ALWC. They also suggested gas-phase oxidation dominate the sulfate and nitrate formation in Lines 623-625. I'm not sure which mechanism the authors were trying to refer to. SO4 can only be efficiently formed through aqueous oxidation of SO2 by O3, which would need aerosol water. Please clarify.

Thanks for the comments and we quite agree that the reaction of $SO_2$ with $O_3$ is quite slow without the participation of aerosol water. Although the observational data shows correlation between sulfate with and or $O_3+NO_2$, but not with ALWC, it doesn't preclude the role of aerosol liquid water in the formation of sulfate. It was simulated by the ISORROPIA model that ALWC reached a moderate level of 24 μg/m$^3$ during P2. Thus, aerosol liquid water should still be an important medium for the aerosol formation. However, based on the data analysis, we think the sulfate pathways in the aqueous phase such as oxidation by $H_2O_2$, catalysis by tracer metals, and oxidation by $NO_2$ should be

overwhelmed by the sulfate pathway of SO$_2$ oxidized by O$_3$.

To make this statement more clearly, the related sentence is revised in Line 518-521 as below.

During P2, both SO$_4^{2-}$ and NO$_3^-$ displayed the most significant correlations with O$_3$ and Ox (O$_3$+NO$_2$), while even negatively correlated with ALWC. In regard of the obvious ozone enhancement phenomenon as discussed in Section 3.3.1, the photochemistry pathway for the secondary aerosol formation (e.g., S(IV) + O$_3$ (aq) → S(VI)) should overwhelm over the aqueous phase pathways, e.g., oxidation by H$_2$O$_2$, catalysis by tracer metals, and oxidation by NO$_2$.

(2) Lines 547- 548, I don't quite understand the calculation in section 3.5. The authors suggested the dust plume in P1 were blown into the oceanic area followed by a mixing period (P2, Lines 260-271). The dust plume in P1 might linger over coastal area. This means the backflow may involve both the dust originally blown out in P1 as well as the dust from Lianyungang. The current calculation of "k" only considers the dust transported from Lianyungang. And the so-called background BC concentration in Pudong cannot represent the BC concentration in the dust plume that lingering over the ocean.

Thanks for the comments. It is a very good point that there is indeed possibility of lingering dust over the ocean that could be also mixed with the dust from Lianyungang during the backflow. The reliability of the developed method depends on the abundances of lingering dust over the ocean. However, there are no observational data over the ocean for the assessment. In this regard, we used observational data in Zhoushan archipelago for additional analysis in the figure below.

[Figure]

The time-series of $PM_{10}$ observed at both Shanghai and Zhoushan are compared. It can be seen that $PM_{10}$ quickly decreased to very low concentrations after the strong dust during P1. This is a common phenomenon that strong dust plumes quickly dissipated due to the strong cold fronts. It should be noted that Zhoushan archipelago is located less than 100km from the coasts. Based on the transport trajectories shown in Figure 2f, the P3 dust plumes were much far away from the coasts. Thus, it could be reasonably conjectured that the lingering dust over the ocean was limited.

However, we do agree with the reviewer that there could be interference from the lingering dust over the ocean. In the revision, we have added a paragraph in Line 639-643 as below.

It should be noted that the simple method devised in this study may have inherent uncertainties. Considering the prolonged duration of the dust event, it is possible that certain dust particles lingered over the open ocean. Consequently, the contributions attributed to aerosol transport should be considered as a conservative estimate or lower bound, rather than an exhaustive assessment.